

# Testing new physics models with global comparisons to collider measurements: the Contur toolkit

A. Buckley[1], J. M. Butterworth[2], L. Corpe[2*], M. Habedank[3], D. Huang[2],
D. Yallup[2†], M. M. Altakach[2], G. Bassman[2], I. Lagwankar[4],
J. Rocamonde[2], H. Saunders[2‡], B. Waugh[2] and G. Zilgalvis[2]

**1** School of Physics & Astronomy, University of Glasgow,
University Place, G12 8QQ, Glasgow, UK
**2** Department of Physics & Astronomy, University College London,
Gower St., WC1E 6BT, London, UK
**3** Department of Physics, Humboldt University, Berlin, Germany
**4** Department of Computer Science and Engineering, PES University, Bangalore, India

* Now at CERN
† Now at University of Cambridge
‡ Now at Tessella

## Abstract

Measurements at particle collider experiments, even if primarily aimed at understanding Standard Model processes, can have a high degree of model independence, and implicitly contain information about potential contributions from physics beyond the Standard Model. The CONTUR package allows users to benefit from the hundreds of measurements preserved in the RIVET library to test new models against the bank of LHC measurements to date. This method has proven to be very effective in several recent publications from the CONTUR team, but ultimately, for this approach to be successful, the authors believe that the CONTUR tool needs to be accessible to the wider high energy physics community. As such, this manual accompanies the first user-facing version: CONTUR v2. It describes the design choices that have been made, as well as detailing pitfalls and common issues to avoid. The authors hope that with the help of this documentation, external groups will be able to run their own CONTUR studies, for example when proposing a new model, or pitching a new search.



# 1  Introduction

The discovery of the Higgs boson was the capstone of decades of research, and cemented the validity of the Standard Model (SM) as our best understanding so far of the building blocks of the universe. The SM boasts a predictive track record worthy of its position as one of the triumphs of modern science. It led to the discovery of the vector bosons $W$ and $Z$, the top quark, and the Higgs boson, and SM cross-section predictions — ranging across ten orders of magnitude from the inclusive jet cross-section at $O(10^{11})$ pb, to electroweak $VVjj$ processes at $O(10^{-3})$ pb — have been found to agree with experimental data through decades of scrutiny, with no significant deviations.

Despite this monumental achievement, the SM is ostensibly an approximation. Qualitative phenomena such as the cosmological matter-antimatter asymmetry, and astrophysical observations consistent with dark-matter and dark-energy contributions to cosmic structure and dynamics suggest directly that the SM is not the whole story. These indications are reinforced by technical issues within the SM such as the "unnatural" need for fine-tuning of its key parameters, and its formal incompatibility with relativistic gravity.

With the absence so far of evidence for electroweak-scale supersymmetry, or of obvious new resonances in measured spectra, the field of collider physics finds itself at a crossroads. For the first time in fifty years, there is no single guiding theory to motivate discoveries. On the other hand, the LHC has delivered the largest dataset ever collected in particle physics, with the promise of a dataset an order of magnitude larger to be delivered by the high-luminosity (HL) LHC in the coming years. A transition from a top-down, theory-driven approach to a bottom-up, data-driven one is needed if we are to use these data to achieve the widest possible coverage of possible extensions to the SM.

The problem is that the field of particle physics does not currently work efficiently in data-driven mode. Searches may take years to produce and concentrate only on certain signatures of a handful of models at a time. These models may even already be excluded, since the new particles and interactions which they feature would have modified well-understood and measured SM spectra. What if we could harness the power of the hundreds of existing LHC measurements preserved in RIVET [1], to rapidly tell whether a model is already excluded? A more comprehensive approach to ruling out models could liberate person-power and resources to focus on the trickiest signatures. This is the purpose of Constraints On New Theories Using RIVET (CONTUR), a project first described in Ref. [2].

The CONTUR method has proven an effective and complementary approach to ruling out new physics models in a series of case studies [3–6], as well as in providing a "due diligence" check for newly proposed models [7,8]. Running a CONTUR-like scan of any newly proposed new-physics model, or when a new search is being designed, should be routine in experimental particle physics, and would potentially liberate search teams to focus on models which have *not* already been ruled out. This shortcut around models which — no matter how theoretically elegant — are already incompatible with model-independent observations will accelerate the feedback loop between theorists and experimentalists, and bring us more efficiently to the long-sought understanding of what lies beyond the SM.

The CONTUR code is now mature enough to turn this vision into a reality, and this manual is intended to accompany the first major user-facing release of the CONTUR code (CONTUR v2, tagged on Zenodo as Ref. [9]), so that theorists and experimentalists who are not CONTUR developers can use this technology to test new models themselves. The CONTUR homepage [10] provides links to source code as well as up-to-date installation and setup instructions.

## 2 Overview

This document is structured as follows: this section gives a general introduction to the CONTUR workflow and design philosophy. Section 3 deals with the relationship between RIVET and CONTUR, and how RIVET analyses in the CONTUR database are classified into orthogonal pools, with advice on adding new analyses. Section 4 runs through setting up and running CONTUR scans over a set of parameter points in a given model. Section 5 explains how CONTUR builds a likelihood function to perform the statistical analysis of the results, and how exclusion values are calculated and analysed. Section 6 takes the user through the various plotting and visualisation tools which come with CONTUR, to help validate and digest results of a scan. Finally, Section 7 concludes the manual. Some of the explanations and figures in these sections have been adapted from a PhD thesis partially focused on the development of CONTUR [11].

Several appendices are provided to give further detail on some functionality, as well as detailed examples. Appendix A provides a detailed flowchart which covers almost all aspects of the CONTUR package described in this manual. Appendix B provides the user with a complete didactic example of the analysis of a beyond-the-SM (BSM) model with CONTUR, using the HERWIG [12] event generator. Appendix C provides detailed descriptions of the various helper executables and other utilities which are provided in the CONTUR package, including details about CONTUR DOCKER containers. Appendix D gives further details about the UFO [13] format, which is used to encapsulate the details of BSM models, while Appendix E details CONTUR compatibility with the SLHA [14,15] format. Appendix F documents how model parameter values can be provided to CONTUR via PANDAS `DataFrame` [16,17] objects. Appendix G provides further details about how to use generators other than HERWIG with CONTUR. Finally, Appendix H provides further detail about the various databases and classifications which are used in the CONTUR workflow.

## 2.1 The CONTUR workflow

The basic premise of CONTUR is that modifications to the SM Lagrangian typically introduce changes to already well-understood and measured differential cross-sections. Therefore, if adding a beyond-SM component to the Lagrangian, *i.e.* a new interaction involving either SM or new BSM fields, would change a measured distribution beyond its experimental uncertainties, then, in simple terms "we'd already have seen it". This can be quantified more precisely in terms of statistical limit-setting, but the upshot is that if one can predict how a given BSM model would modify the hundreds of observables measured in existing LHC measurements, then it is already possible to exclude regions of its parameter space without the need for a dedicated search.

This perspective turns the immediate model-testing challenge from an experimental one into a computational and book-keeping one. Can we design a workflow to take a BSM model with a set of parameter values, generate simulated events from it, quickly infer the effect of those events in each bin of the LHC measurements to date, and compute the $p$-value (and hence exclusion status at some confidence level) for that model point? Can one then efficiently repeat that procedure over a range of parameter points, to determine the regions of parameter space which are excluded? CONTUR is a tool that implements such a process. It builds on several existing data formats, conventions and packages to achieve this goal, and automatically handles the steering of model parameters and associated book-keeping on the user's behalf. The basic workflow is illustrated schematically in Figure 1, and in much more detail in Figure 5 of Appendix A

The first requirement is that the BSM model be implemented in a Monte Carlo event generator (MCEG) such that its parameters can be set, and simulated events generated for analysis. Historically this required manual coding, and hence focused on BSM models such as supersymmetry, technicolor, and new quarks and vector bosons, which were considered leading candidates for new physics before LHC operation. The Super-symmetric (SUSY) Les Houches Accord (SLHA [14, 15]) format was developed as a MCEG-independent way of specifying the mass and decay spectra of such models, and is understood by many MCEGs. As the "obvious" BSM models waned and gave way to a much wider spectrum of possibilities, a complementary format — the Universal FeynRules Output [13] (UFO) — was developed to transmit not just parameter choices but the entire model, built up from a Python-based encoding of the BSM Lagrangian. The combination of UFO and SLHA files provides an industry-standard way to package the details of any BSM model, such that most MCEGs can interpret it without needing model-specific code. Its ubiquity means that theorists routinely publish UFO files when proposing a new model, making them easy to study and test. Details on how to use a new UFO file as an input to CONTUR can be found in Appendix D, and use of SLHA-driven configurations in Appendix E.

MCEGs use the specified BSM model and parameters to simulate new-physics events in high energy collisions. In the default CONTUR workflow, the HERWIG [12] event generator is used (see Appendix B for an example), but other event generators, such as MADGRAPH5_AMC@NLO [18] and POWHEG [19] are also supported. Additionally, if events are already generated and parameter steering is therefore not required, RIVET and thus CONTUR can analyse events stored in HEPMC [20, 21] format. More details on support for various event generators in CONTUR are given in Appendix G.

The generated events are fed into RIVET (see Section 3), the output of which then corresponds to the extra BSM contribution which would have been present in any of the hundreds of spectra measured at the LHC so far, if the generated model existed in nature. The BSM component can then be compared to the size of the uncertainty for the measurement, and optionally to the SM expectation. Measurements are grouped into orthogonal pools (see Section 3.1), and CONTUR uses the best constraint from each pool to form a global exclusion measure for a

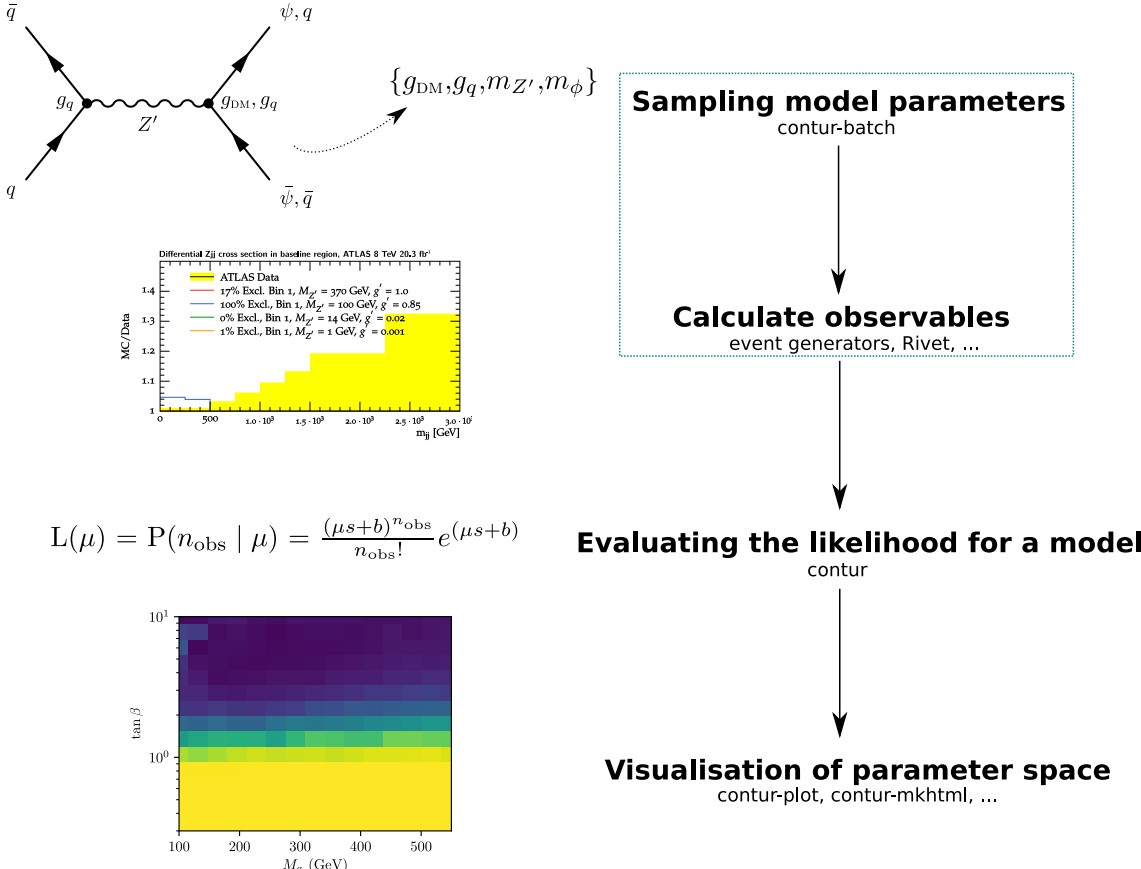

Figure 1: A simplified schematic of the CONTUR workflow. The dotted box denotes the portion of the workflow that makes extensive use of external packages, affording multiple options, such as the choice of MCEG. These two steps are described together in Section 4, 'Sampling model parameters'. The next stage taking physics observables as inputs to a statistical analysis, 'Evaluating the likelihood for the model' is described in Section 5. Finally some of the tools to visualise the output of the likelihood analysis, 'Visualisation of parameter space', are covered in Section 6. A much more detailed diagram is available in Figure 5 of Appendix A.

given model at a given set of parameter values. The details of the statistical treatment can be found in Section 5.1.

This whole process typically takes under an hour for a single point on a single compute node. Repeated for a grid of parameter values, and running in parallel on a compute farm, CONTUR can determine in a few hours whether wide regions of a model's parameter space are still potentially viable, or already excluded by existing LHC measurements. The CONTUR package comes with plotting and visualisation tools to present and digest the results of a scan. These are discussed in Section 6 and Appendix C.

## 2.2 The CONTUR philosophy

CONTUR is designed to efficiently address the question *"How compatible is a proposed physics model with published LHC results?"* This question needs to be asked each time a new model is proposed. The ability to answer it depends on a number of factors.

Firstly one must define what is meant by "LHC results". Collider physics experiments

produce a variety of different types of results, which can be broadly classified as follows.

1. Extraction of fundamental parameters of the SM, such as the $W$ mass, the Weinberg angle, etc... Such results give experimental constraints on SM parameters which are calculable analytically in perturbative field theory.

2. Extraction of so-called inclusive quantities, such as (for example) the total production cross-section $t\bar{t}$, or $WW$. This usually involves theory input to extrapolate into regions outside the acceptance of the measurement.

3. Measurements of fiducial *particle-level* observables. In other words, observables corrected for detector effects or "unfolded", but not extrapolated beyond detector acceptance. Comparing predictions to such measurements requires the generation of simulated events, making use of perturbative field theory but also non-perturbative models, and numerical MC techniques, so that fiducial phase space selections may be applied to final-state particles.

4. Measurement of *detector-level* distributions. This is the most common type of result used in searches by ATLAS and CMS. They are faster to produce than unfolded results, since the step of validating the model-independence of the unfolding (but not of calibration) can be skipped. However they cannot be compared to theory without an additional detector simulation step.

5. Exclusion regions from searches, usually derived from the detector-level distributions mentioned above. These can sometimes be reinterpreted in terms of new models, but may have significant implicit model-dependence.

This categorisation is shown schematically in Figure 2. The direction of the arrow indicates increasing calculational complexity required of the theory to compare the result to SM predictions. At first, just an analytical calculation of the SM parameter is needed. Then, MC simulation at parton and particle-level are required. Finally, the effects of the detector must be modelled. The level of model assumption built into the experimental data increases in the opposite direction.

All interpretations, or re-interpretations, of results involve compromises and approximations. The CONTUR philosophy is to strive for speed and coverage of new models, at the expense of some precision and sensitivity. To do this, we focus primarily on fiducial, particle-level measurements, as a compromise between model dependence and detector independence: that is, minimal theory extrapolation in the measurement, and minimal detector dependence in the BSM predictions. This means using results of type 3, and in some circumstances 4, from the list above. A general discussion of reinterpretation tools and requirements is given in Ref. [22].

In addition to making use of particle-level measurements to help exclude new physics models, another pillar of the CONTUR philosophy is to use inclusive event generation instead of exclusively generating individual processes. Inclusive event generation has the advantage of covering all allowed final states which would be affected if that BSM model were realised. Generating events in this way, CONTUR can paint a more comprehensive picture of the exclusion across all manner of final states, rather than focusing on the most spectacular signatures of a new model. Indeed, there are several cases in recent CONTUR papers where exclusion power for a model in some region of parameter space has come from an unexpected signature, which might not have been tested if the user had to actively switch on individual processes. By contrast, determining which processes are most important in different regions of model parameter space is not trivial if one is not an expert in the phenomenology of a particular BSM model. HERWIG is an event generator which features an inclusive mode which generates all $2 \rightarrow 2$ processes

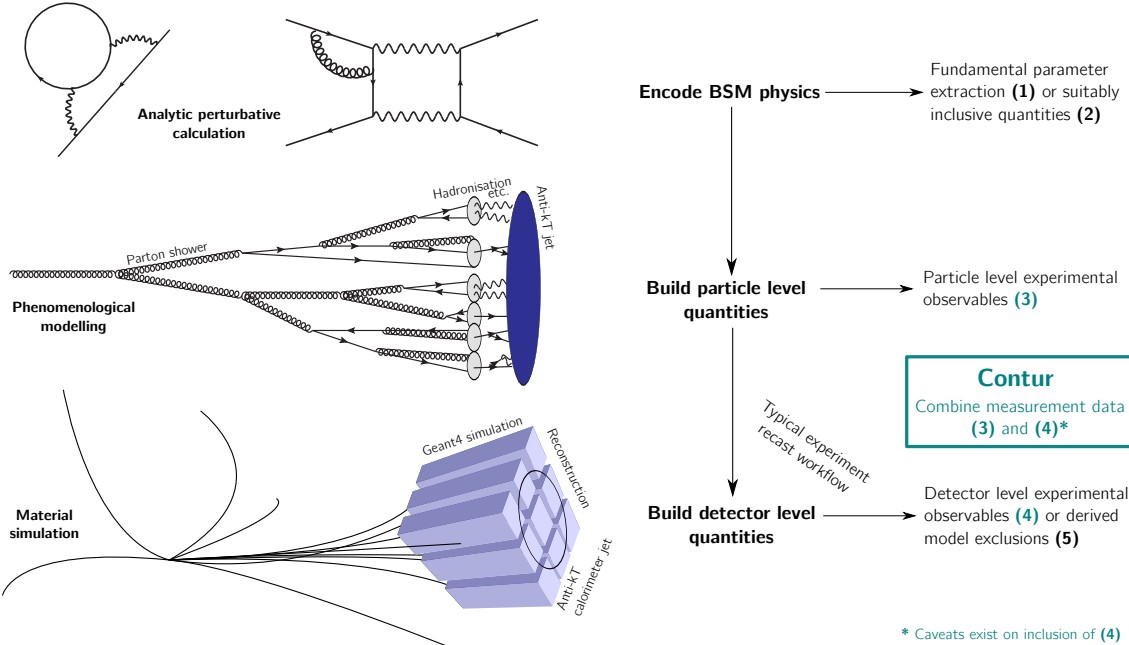

Figure 2: A schematic illustrating the levels at which data and theory may be compared in LHC physics. The vertical (downward) arrows show the direction of increasing complexity of the theoretical prediction; in the reverse direction, increasingly complex corrections must be applied to the data. Horizontal arrows show the comparison data available at each level.

featuring a BSM particle in the propagator or an outgoing leg. For this reason, HERWIG is the default event generator used in CONTUR, as can be seen in the example in Appendix B. Nevertheless, CONTUR retains the possibility to study individual processes, for instance, when only a particular process is of interest, or to check how much of a contribution would come from processes which are more complex than $2 \rightarrow 2$.

## 2.3 Limitations of CONTUR

It is important to note that CONTUR is not at present a 'discovery' tool. It will not identify regions of BSM parameter space which are more favoured than the SM; such regions will show up as 'allowed', but the test statistic is one-sided and gives no more positive information about a BSM scenario than that. In any case, CONTUR only uses data which have been shown to agree with the SM.

Most of the other limitations of the CONTUR method at present stem from incomplete information published by the experiments. Three common issues arise: the SM prediction of a measurement is not published, the bin-to-bin correlation information for the systematic uncertainties is not made available, or the public information contains hidden, model-dependent assumptions. These items are discussed in more detail in the following.

The fact that most entries in the HEPDATA library (described in Section 3) currently do not include a SM prediction means that assumptions must be made with respect to the null hypothesis in the CONTUR method. In particular, if the SM-prediction information is not available, CONTUR assumes the data are identically equal to the SM. This is an assumption that is reasonable for distributions where the uncertainties on the SM prediction are not larger than the uncertainties on the data; it is also the assumption made in the control regions of many searches, where the background evaluation is "data-driven". When used in this mode,

CONTUR would be blind to a signal arising as the cumulative effect of a number of statistically insignificant deviations across a range of experimental measurements[1]. To extract such a signal properly requires evaluation of the theoretical uncertainties on the SM predictions for each channel. These predictions and uncertainties are gradually being added to CONTUR and can be tried out using a command-line option (see Ref. [6] for a first demonstration). For these reasons, limits derived by CONTUR where the theory predictions are not used directly are best described as expected limits, delineating regions where the measurements are sensitive and deviations are disfavoured. In regions where the confidence level is high, they do represent a real exclusion.

A further limitation comes from a lack of information about correlations between bins in some published measurements. For measurements which are not statistically limited, systematic correlations between bins may be important. Without knowing the size of the correlations between bins, CONTUR must only use the single most sensitive bin in a given distribution to avoid double-counting correlated excesses across multiple bins. This limits the sensitivity of CONTUR when a BSM signal is spread over several bins in a distribution. However, in an increasing number of cases, a breakdown of the size of each major source of correlated uncertainty in each bin is provided by the experiments, and in these cases CONTUR is able to make use of them.

More fundamentally, some measurements are defined in ways which make their use in CONTUR limited or impossible. This usually occurs because SM assumptions have been built into the measurement (for example extrapolations to parton level, or into significant unmeasured phase space regions), important selection cuts (a common example being jet vetoes) have not been implemented in the fiducial phase space definition, or large data-driven background subtractions (for example in $H \to \gamma\gamma$) have been made. Existing examples, and the conditions under which some such routines may or may not be used, are discussed in Section 3.3.

Finally, and trivially, if a RIVET routine and HEPDATA entry are not available for a measurement, CONTUR cannot use it.

## 2.4 Code structure and setup

The CONTUR tool is mostly structured in the standard Python form, with a package directory `contur` containing the majority of processing logic, and secondary `bin` and `data` directories respectively containing executable "user-facing" scripts, and various types of supporting data files. A set of unit tests is implemented in the `tests` directory, using the PYTEST framework. An outline of the directory structure can be seen in Listing 1.

The `contur` Python package is internally divided into modules reflecting the distinct parameter-scanning, MC-run management, and statistical post-processing tasks of CONTUR operation. The `scan` module provides helper functions for generating and combining parameter-space scan data, `plot` implements the standard data-presentation formats, and the `run` module provides the logical cores of the main user scripts in a form amenable to PYTEST testing. The statistical machinery central to CONTUR lives in the main `contur` namespace, supported by utility functions, data-loading tools, and CONTUR's analysis-pool data from the `util`, `factories` and `data` modules. These classes are documented in inline documentation using PYDOC, which is linked from the main CONTUR repository.

---

[1]This would be particularly worrisome in low-statistics regions, where outlying events in the tails of the data will not lead to a weakening of the limit, as would be the case in a search. However, measurements unfolded to the particle-level are typically performed in bins with a requirement of minimum number of events in any given bin, reducing the impact of this effect (and also weakening the exclusion limits).

Listing 1: An illustration of CONTUR software file structure.

```
bin                          User-facing executables and scripts
    contur                   The main CONTUR executable
    contur-batch             A utility for submitting CONTUR scans to HPC systems
    contur-plot              A utility for plotting CONTUR results
    contur-export            A utility for exporting CONTUR results to a CSV file
    ...
contur                       Main Python package containing internal processing logic
    config                   Configuration and initialisation logic
    data                     Data manipulation and database-management code
    factories                Internal tools to produce scan and likelihood objects
    plot                     Plotting and visualisation code
    run                      Logical core of main CONTUR scripts
    scan                     Tools for parameter-space scan data manipulation
    util                     General-purpose utilities and helper functions
    ...
contur-visualiser            An auxiliary utility for interactive visualisation
    ...
data                         Directory containing supporting data files
    DB                       SQL files for the database of available RIVET analyses
    Models                   A library of example BSM UFO models
    Rivet                    New or overloaded RIVET analysis sources and files
    Theory                   Additional SM theory predictions for RIVET analyses
    TheoryRaw                Files from which SM theory predictions were imported
    ...
docker                       Directory containing build files for DOCKER containers
    ...
tests                        Directory encapsulating the unit test framework
    ...
```

The `bin` directory contains the main user scripts described in this paper, plus the auxiliary ones described in Section C. The `data` directory contains a mixture of bundled and generated data. Included in the release are:

- sets of model files and generation templates created so far,

- any modified or new RIVET analysis codes and reference data not bundled with the RIVET release,

- theory-based background estimates for analyses where the data need not be assumed to be purely SM.

Files generated by the user after installation include:

- the compiled RIVET analysis-plugin libraries,

- analysis-pool database (see Appendix H),

- MC-run template files.

The CONTUR package relies on a compiled RIVET installation and set of analysis overrides, and requires manually copying template files from `data/Models` to run MC scans. For these reasons, it is not usually recommended to perform the installation using the Python SETUPTOOLS

scheme [2]. Installing and using CONTUR is instead normally done directly from the downloaded project directory, by sourcing a script called `setupContur.sh` on each shell session, and by running the `make` command only once upon installation. Sourcing `setupContur.sh` sets environment variables (such as `CONTUR_ROOT`) that RIVET uses to locate custom analyses and data. Additionally, this script appends the CONTUR Python module path and the executable `bin` directory to the system `PYTHONPATH` and `PATH` respectively, mirroring the function of Python's standard SETUPTOOLS. As the CONTUR package makes use of a compiled database, referencing analysis lists derived from this at run time, setting these environment variables is needed to operate parts of the workflow. Furthermore, a short python script is run when `setupContur.sh` is sourced which checks the various dependencies and paths.

## 3  RIVET analyses

RIVET functions as a library of preserved particle-level measurements from colliders. Each publication has a corresponding RIVET routine: a runnable C++ code snippet which encapsulates at particle-level the definition of the measured cross section. RIVET routines can be thought of as filters that select generated events which would enter the fiducial region, and project their properties into histograms with the same observables and binnings as the measurements. Several hundred measurements are preserved in this way, many of them from LHC experiments. The output of RIVET is a set of statistical analysis objects stored in the native YODA format. YODA files are human-readable text files with structures to encode binnings, statistical moments and other (correlated) uncertainties for the usual statistical analysis objects used in HEP: one- and two-dimensional histograms, one- to three-dimensional scatter plots, profile histograms, and so on...

The HEPDATA [23] repository contains a digitized record of the measured cross-section values and their uncertainties, sometimes also including the best SM theory predictions at the time, and sometimes including a breakdown of uncertainties in each bin or other correlation information. This information about experimental measurements from HEPDATA can also be exported in the YODA format. YODA files are synchronised between RIVET and HEPDATA whenever a new release of RIVET is made, so that a faithful comparison of generator output to measured data and uncertainties can be made.

The measurements present in RIVET and used by CONTUR in the current version are those in Refs. [24–126] although new measurements are continually being added. CONTUR re-uses these encapsulated analysis routines, but runs with generated BSM events rather than the SM process which was typically the target of the measurement. RIVET is specifically designed to run multiple (or indeed, all) analysis plugins simultaneously for a given beam configuration. It has been optimised to do this quickly and efficiently. Thus BSM events generated by HERWIG or another MCEG are filtered through all available plugins, leading to a multitude of histograms showing if, and where, the signal would have appeared in existing LHC measurements. The size of the signal can then be compared to the relevant HEPDATA reference histogram, to decide if the set of BSM parameters in question would have produced a distortion to the SM spectrum beyond measured uncertainties. This simple stacking of BSM contributions onto existing measurements needs additional information for certain measurement types. Indeed, normalised histograms are complemented with a fiducial cross-section factor in the analysis database (when this is provided by the experiment), which allows rescaling to the differential

---

[2]The repository does contain a `setup.py` file that allows installing CONTUR as a package, for example to allow the statistical interpretation function to be accessible from other programmes. However, most of the CONTUR functionality will not be available with this method, which is currently a work in progress, and for most users it is not recommended.

cross-section for addition, then re-normalisation. Ratio plots have a similar special treatment, and a profile histogram treatment is being developed in the same way.

## 3.1 Categorisation of RIVET routines into orthogonal pools

If injection of BSM signal leads to an excess in a measured distribution, there may also be excesses in measurements of similar final states produced from partially overlapping datasets, leading to correlations. Since correlations between the measurements cannot be accounted for, this could lead to an overestimate in the sensitivity. To avoid such double counting, CONTUR classifies RIVET histograms into orthogonal pools based on centre-of-mass energy of the LHC beam, the experiment which performed the measurement, and the final state which was probed. For analyses which measured several final states which are implemented as different options within the RIVET plugin, it is possible to sort histograms from the same analysis into different pools. If there are orthogonal phase space regions measured within the same analysis (for example different rapidity regions in a jet cross section) it is possible to combine the non-overlapping histograms into a "subpool", in which case the combined exclusion of the subpool will be evaluated, and treated as though it came from a single histogram.

The results from each pool can then be combined without risk of over-stating the sensitivity to a given signal. Analysis pools are named as ⟨*Experiment*⟩_⟨*Center-of-mass energy*⟩_⟨*Final state*⟩, where:

- ⟨*Experiment*⟩ can be `ATLAS`, `LHCB` or `CMS` at present;

- ⟨*Center-of-mass energy*⟩ can be 7, 8 or 13 TeV;

- ⟨*Final state*⟩ is a short string which loosely describes the final state, with details given in Table 1;

These pools, and other information, are stored in a database, described in Appendix H. Although CONTUR presently only uses LHC results, measurements from non-LHC experiments, such as LEP or HERA, could also in principle be included provided they were made in a model-independent way, and preserved in an appropriate format with a RIVET routine and HEPDATA entry. All that would be needed is to add additional beam modes to simulate collisions of the appropriate particles at appropriate energies.

## 3.2 Adding user-provided or modified RIVET analyses

To make a local modification to an existing RIVET routine or include one not yet in the RIVET release, the new or modified analysis plugin can be copied into the `contur/data/Rivet` directory along with any updated reference files. Further, if a new theory calculation becomes available this can be added to the `contur/data/Theory` directory. The new routine can be compiled with a simple `make` call, followed by `setupContur.sh`. This will over-ride the default (un-modified) version of the replaced analysis for the next run. The new analysis should also be added to the `analysis.sql` file, documented in Appendix H.

## 3.3 Rivet routine special cases and common pitfalls

Most analyses preserved in RIVET are particle-level measurements, meaning the measurement is to a large extent defined in terms of an observable final state, and the effects of the detector have already been corrected for during the unfolding procedure, within some fiducial region. As a result, predictions and measurements can be compared directly, without the need for smearing or detector simulation. Some exceptions and caveats exist however, limiting the applicability of some analyses. The current known special cases are discussed below, and their categorisation in the CONTUR database structure is discussed in Appendix H.2.

Table 1: Description of the currently considered ⟨*Final state*⟩ tags used to sort analysis histograms into orthogonal pools.

| ⟨*Final state*⟩ tag | Description of target final state |
|---:|---|
| 3L | Three leptons |
| 4L | Four leptons |
| EEJET | $e^+e^-$ at the $Z$ pole, plus optional jets |
| EE_GAMMA | $e^+e^-$ plus photon(s) |
| EMETJET | Electron, missing transverse momentum, plus optional jets (typically $W$, semi-leptonic $t\bar{t}$ analyses) |
| EMET_GAMMA | Electron, missing transverse momentum, plus photon |
| GAMMA | Inclusive (multi)photons |
| GAMMA_MET | Photon plus missing transverse momentum |
| HMDY | Dileptons above the $Z$ pole |
| HMDY_EL | Dileptons above the $Z$ pole, electron channel |
| HMDY_MU | Dileptons above the $Z$ pole, muon channel |
| JETS | Inclusive hadronic final states |
| LLJET | Dileptons (electrons or muons) at the $Z$ pole, plus optional jets |
| LL_GAMMA | Dilepton (electrons or muons) plus a photon |
| LMDY | Dileptons below the $Z$ pole |
| LMETJET | Lepton, missing transverse momentum, plus optional jets (typically $W$, semi-leptonic $t\bar{t}$ analyses) |
| METJET | Missing transverse momentum plus jets |
| MMETJET | Muon, missing transverse momentum, plus optional jets (typically $W$, semi-leptonic $t\bar{t}$ analyses) |
| MMET_GAMMA | Muon, missing transverse momentum, plus photon |
| MMJET | $\mu^+\mu^-$ at the $Z$ pole, plus optional jets |
| MM_GAMMA | $\mu^+\mu^-$ plus photon(s) |
| TTHAD | Fully hadronic top events |
| L1L2MET | Different-flavour dileptons plus missing transverse momentum (*i.e.* $WW$ and $t\bar{t}$ measurements) |

**Ratio plots**  The current most powerful particle-level measurement of missing energy is in the form of the measurement of a ratio of $\ell\ell$ plus jets to missing energy plus jets [92]. The cancellations involved bring greater precision, but the SM leptonic process is hard-coded as the denominator, so the results are not reliable for models that would change this — for example, enhanced $Z$ production will contribute to both the numerator and the denominator. For models where this is expected to be an issue, the analysis may be excluded by setting the `--xr` flag at CONTUR runtime.

**$H \rightarrow \gamma\gamma$**  These fiducial measurements [119] are very powerful for models which enhance SM Higgs production. However, they rely on a fit to the $\gamma\gamma$ mass continuum to subtract background. Signals from models which enhance non-resonant $\gamma\gamma$ production would presumably have influenced this fit, and might have been absorbed into it, so looking at their contribution only in the $H$ mass window will overestimate the sensitivity. These analyses may be excluded in such cases by setting the `--xhg` flag at CONTUR run time.

**Searches** Detector-level RIVET routines do exist for some searches, and can be used by CONTUR [41, 74]. In this case RIVET's custom smearing functionality is used, and the SM background from HEPDATA is used for comparison. These searches may be turned on by setting the -s flag at CONTUR run-time.

**$H \rightarrow WW$** Like $H \rightarrow \gamma\gamma$, these measurements [46, 117] could potentially be very important when SM Higgs production is enhanced. However they involve very large data-driven background subtraction (principally for top), and the reliability of this for non-SM production mechanisms (of Higgs, $WW$, or just dileptons and missing energy) is in general hard to determine. These analyses may be turned on by setting the --whw flag at CONTUR run time.

**ATLAS $WZ$** This analysis [32] may be useful for models which enhance $WZ$ production, but it calculates event kinematics using the flavour of the neutrinos, and so its impact on other missing energy signals is difficult to evaluate. The analysis may be turned on by setting the --awz flag at CONTUR run time.

**$b$-jet veto** Analyses targeting $WW$ production processes generally use $b$-jet vetos to suppress $WW$ production via $t\bar{t}$. In some cases, this kinematic requirement is made only at detector level, and not included in the fiducial cross section definition implemented in the RIVET routine [46, 53, 71, 123]. These analyses are therefore likely to give misleading results when used on non-SM $WW$ production processes.

These exceptions are catalogued in the analysis database (see Appendix H) and should be taken into account when implementing or adding a new analysis to CONTUR. Some guidelines for designing analyses to minimise their model dependence and maximise their impact in a CONTUR-like approach are given in Ref. [22]. The most important principle is that theory-based extrapolations should be avoided where possible, both for background subtraction and unmeasured signal regions. This essentially means defining a fiducial measurement region in terms of final state particles which as far as possible faithfully reflects the actual detector-level event selection.

## 4   Sampling model parameters

New physics models usually have a number of parameters which are not fixed. Surveying such a model begins with identifying the parameters of interest and sampling points within that parameter space. CONTUR provides a simple custom tool-set to facilitate this, currently limited to sampling a small number of parameters in a single scan.

The scanning functionality is implemented in the `scan` module, and user interaction is mostly controlled with the `contur-batch` executable. This executable requires three core user-defined components governing the behaviour of the scan:

- A run information directory containing the required common files such as model definition and analysis lists. Preparing this directory is outlined in Section 4.1;

- A parameter card dictating how the parameters of interest should be sampled. The structure of this file is explained in Section 4.2;

- A template generator steering file. This depends on the MCEG being used, and is discussed in Appendices B.5 and G.

Alternatively to constructing scans of model parameters, specific parameter choices can be manually sampled. By calculating observables for a chosen set of parameters (demonstrated using HERWIG in Appendix B.3), a file containing YODA analysis objects can be fed directly to the likelihood machinery described in Section 5. This allows manual sampling of a parameter space using the CONTUR likelihood machinery.

## 4.1 Initial grid setup

The list of observables to calculate is dependent on the available list of compiled RIVET analyses. As discussed in Section 3, this list can be augmented by the user and is subject to change dependent on the RIVET version used. The `contur-mkana` command line utility is called to generate some static lists of available analyses to feed into the MCEG codes. Specifically, HERWIG-style template files are created in a series of `.ana` files, and a shell script to set environment variables containing lists of analyses useful for command-line-steered MCEGs is also written. After `contur-mkana` is invoked, re-sourcing the `setupContur.sh` script defines the necessary environment variables. The HERWIG analysis list files will also now exist in `data/share`. Local model files and, if using HERWIG, the analysis list files, should be copied to a subdirectory of the local run area (default name `RunInfo`)[3]. This subdirectory is then supplied to the main `contur-batch` executable via the `grid` command-line argument.

## 4.2 Parameter card setup

The parameter card is supplied to the `contur-batch` executable via the `--param_file` command-line argument. The structure of this file is based on the input/output structure defined by the Python CONFIGOBJ package. Entries delineated by square braces define dictionaries named as the contained string. Double square braces are a dictionary within the parent dictionary. The two main dictionaries to steer the parameter sampler are `Run` and `Parameters`. An example parameter card with three model parameters is given in Listing 2. Two additional dictionaries are implemented, which allow the user to make processing more efficient by skipping certain points (using a block named `SkippedPoints`)[4] and scaling the number of events generated at each point (using a block named `NEventScalings`), since some points may need to be probed with more precision than others. The `NEventScalings` dictionary is only applied if the grid is submitted using the `--variablePrecision` option of `contur-batch`. Both the `SkippedPoints` and `NEventScalings` dictionaries can be added automatically to a parameter card using the `contur-zoom` utility, which is designed to help the user iteratively refine a parameter scan, and which is documented in Appendix C.6.

### 4.2.1 Parameter card `Run` arguments

The `Run` block is intended to control high level steering of the parameter sampler. Two dictionary keys are defined in this block:

- `generator`, path to a shell script that configures the necessary variables to setup the event generator;

- `contur`, path to a shell script that configures the necessary variables to load the CONTUR package.

Both of these callable scripts are expected to set up the required software stack to execute the calculation of observables on a High Performance Computing (HPC) node.

---

[3]These files will be copied automatically by `contur-batch` if not already present.

[4]In future, we intend to make further use of PANDAS `DataFrame` compatibility to provide such functionality more elegantly.

Listing 2: An example CONTUR configuration file for a model with three free parameters

```
[Run]
generator = "/path/to/generatorSetup.sh"
contur = "/path/to/setupContur.sh"

[Parameters]
[[x0]]
mode = LOG
start = 1.0
stop = 10000.0
number = 15
[[x1]]
mode = CONST
value = 2.0
[[x2]]
mode = REL
form = {x0}/{x1}
```

#### 4.2.2 Parameter card `Parameters` arguments

Within the `Parameters` dictionary, a series of sub-dictionaries (in double square braces) define the treatment of each parameter in the model. The string used as the name of this dictionary is the name of the parameter, and must also appear in the MCEG run-card template. The `mode` field defines the type of the parameter, and opens additional allowed fields modifying its behaviour. The available values for `mode`, with the sub-list detailing the unique additional parameters for each, are given below:

- `CONST`, a constant parameter.

  - `value`, a float with the value to assume for this parameter.

- `LOG/LIN`, a uniform logarithmically- or linearly-spaced parameter.

  - `start/stop`, the floats of the boundaries of the target sampled space for this parameter (note: `start` must be a smaller number than `stop`).
  - `number`, an integer number of values to sample in the range.

- `REL`, a relative parameter, defined with reference to one or more of the other parameters.

  - `form`, any mathematical expression that Python can evaluate using the `eval()` function form of the standard library, where parameter names wrapped by curly braces, as seen on Listing 2, will be replaced by the value of that parameter before evaluating the expression. The name between braces must match exactly that of the parameter as specified in the `Parameter` block. For safety and efficiency, it is preferable (and often necessary) to use the `DATAFRAME` mode if complex mathematical expressions (*i.e.* anything beyond basic arithmetic operations) are required to generate the desired value for this parameter.

- `SINGLE/SCALED`. Single string substitution. If the parameter name is "`slha_file`", provide a path to a single SLHA file as `name` which will be treated as described in Appendix E.

Listing 3: Snippet of a HERWIG input card for the same three free parameters as previously defined in Listing 2.

```
read FRModel.model
set /Herwig/FRModel/Particles/X:NominalMass {x0}
set /Herwig/FRModel/FRModel:x1 {x1}
set /Herwig/FRModel/FRModel:x2 {x2}

insert HPConstructor:Incoming 0 /Herwig/Particles/u
insert HPConstructor:Incoming 0 /Herwig/Particles/ubar
insert HPConstructor:Outgoing 0 /Herwig/FRModel/Particles/X
set HPConstructor:Processes SingleParticleInclusive
```

- DIR, if using the SLHA specification, giving a directory to the SLHA files as `name`. Each file in the directory will generate a separate run point with the parameters set accordingly.

- DATAFRAME, one can also provide a PANDAS `DataFrame` in a PICKLE file as `name`, which provides the parameters to vary and their values, one point for each row of the table. PANDAS `DataFrame` support is further documented in Appendix F .

With these tools, many parameters available in the model can be scoped in the CONTUR parameter sampler. The parameters whose `mode` is either LOG, LIN or DATAFRAME are the scanned parameters, and the number of such parameters is the dimensionality of the scan. REL or CONST parameters are then ways to correctly set the additional parameters of the model. Any dimension of scan is technically possible but typically only up to two or three parameters have been considered in physics studies using CONTUR. For high-dimensional scans, `contur-export` allows exporting results to a CSV file, so that alternative visualisation tools beyond `contur-plot` can be used (see Appendix C.2).

The CONTUR scanning machinery could in principle be extended to find the least constrained parameter point: this could be done by making use of the CONTUR code interface and connecting to a numerical scanner or minimiser. This would require more efficient scanning of multi-dimensional parameter spaces: this is an area of active research in the CONTUR team and in the reinterpretation community more widely.

## 4.3 Generator template

To interface an MCEG with the CONTUR parameter sampler, a template of the generator input has to be provided. This template file is supplied to the `contur-batch` executable via the `--template_file` command line argument. The parameters that are scoped in CONTUR as described in Section 4 are then substituted into this file, thus defining the generator run conditions.

Following the example of Listing 2 for a CONTUR parameter file, a snippet of the matching HERWIG input card is shown in Listing 3. Much of the syntax is HERWIG-specific, and further discussion is left to the HERWIG documentation. Important features to notice are that the parameters are being defined in the HERWIG FRModel (short for FEYNRULES model, the placeholder for a HERWIG-parsed UFO model file). The names within curly braces match the parameter dictionary names in the parameter card file, allowing numeric values for each to be substituted in, following the rules defined in Section 4.2. Since this workflow is based upon string parsing and substitution, any event generator configuration that can be steered in a similar way can be substituted. In the example, the mass of the $X$ particle has been scanned with the CONTUR sampler by varying the defined parameter, x0.

Listing 4: An example of the CONTUR formatted grid directory for a single beam energy and single point scan.

```
myscan00                        Parent scan directory
    13TeV                       Subdirectory for each requested beam energy
        sampled_points.dat      A flat file, listing of all the sampled points
        0000                    Directory for each sampled point
            params.dat          Flat file with the chosen parameter point
            LHC.in              Generator template with substituted parameter values
            runpoint_000.sh     Shell script to execute
```

An example of the process definition is also included in Listing 3 for this toy model. In this example, the instruction given to the generator is to inclusively generate all $2 \rightarrow 2$ processes with incoming up and anti-up quarks, and an outgoing hypothesised $X$ particle. According to the Feynman rules in the parsed FRModel.model file, all allowed diagrams will be generated. This is the ideal generator running mode for CONTUR, consistent with its inclusive philosophy. However, not all generators provide this inclusive option; also, in some cases it may be useful to focus on specific processes.

As motivated in Section 2.2, this generator setup should be set to generate signal-only contributions to the relevant observables. The statistical analysis detailed in Section 5 will treat the observables resulting from generator runs as being additive signal contributions to the background model.

Specific worked examples of setting up the generator template for the default HERWIG event generator chain are given in Appendix B. Support for MADGRAPH and POWHEG workflows is also implemented and examples are presented in Appendix G.1 and Appendix G.2 respectively. The choice of generator is controlled by the --mceg command-line variable, defaulting to HERWIG.

## 4.4 Grid structure and HPC support

The execution of observable calculation in CONTUR is realised in two steps: definition of the event-generation and observable-construction jobs, and execution of those jobs.

First, if .ana files are required and do not already exist locally, they will be automatically copied by contur-batch from $CONTUR_ROOT/data/share to the local RunInfo directory. Next, a run directory will be created (named myscan## by default), with a subdirectory for each distinct set of run conditions (currently the three available LHC beam energies). In a dedicated subdirectory of each of these for each point in the parameter grid, the sampler creates all associated generator files, with the required commands to run the generator and the selected RIVET analyses. A shell script containing all the commands to execute the generator run from a fresh login shell is also written. An example scan directory is shown in Listing 4.

Next, the scripts which perform the calculations for each parameter point need to be executed. The contur-batch executable will automatically send each job to a HPC node. CONTUR supports the PBS, HTCONDOR and SLURM batch systems, the one in current use being controlled by use of the the --batch command-line argument. The default behaviour is to use PBS submission, where the queue name is controlled by the queue command-line argument. SLURM differs from PBS only in use of the sbatch command-line tool in place of qsub, while the HTCONDOR system differs from the others in not having queues and having to generate a job description file (JDF) for each scan point's condor_batch call.

Alternatively, if the --scan-only command-line option is used, contur-batch will only generate the batch scripts (and JDFs if necessary) but not submit them, leaving detailed run

control entirely to the user. In either mode, no batch-system management is performed by CONTUR once the jobs are running: for this you should use the suite of tools specific to your batch system (qstat, qdel, etc., or their SLURM or HTCONDOR equivalents).

The contur-batch executable also controls the number of events which are generated for each parameter point (using the --numevents option, defaulting to 30,000) [5]. During execution, once the generator has reached the requested number of events, the observables calculated by RIVET are stored as filled histograms in the YODA histogram format. Each parameter space point subdirectory in the grid as shown in Listing 4 will have a corresponding YODA file containing the calculated observables.

## 5 Evaluating the likelihood for a model

Calculation of the CL$_s$ exclusion at a given point in parameter space requires the construction of a likelihood function for that point. The main analysis executable, called simply contur, is responsible for this task. Taking as input a series of calculated observables in YODA format, this can be run either on a single point in parameter space, or on a grid of points generated using contur-batch, in which case a map of the likelihood of the parameter points explored by the parameter sampler is constructed. This section describes the calculation of the likelihood for an individual point in parameter space.

As this is the main analysis component in CONTUR, the functionality is implemented throughout the package modules. The core analysis classes are implemented in the factories module. The entry-point analysis class is named Depot which should contain the majority of the relevant user-access methods. Several intermediate classes handle various aspects of the data flow, down to the lowest-level class defining the statistical method, Likelihood. The data module implements much of the interaction between RIVET and CONTUR, defining how to build covariance matrices for example. The run module implements the behaviour of the executable, and interaction of this calculation with the rest of the modules.

### 5.1 Statistical method

A test statistic based on the profiled log-likelihood ratio can be written,

$$ t_\mu = -2\ln\lambda(\mu) = -2\ln\frac{L(\mu, \hat{\hat{\nu}}(\mu))}{L(\hat{\mu}, \hat{\nu}(\hat{\mu}))}, \tag{1} $$

with $\mu$ being the parameter of interest (POI) and $\nu$ being nuisance parameters. A single hat, e.g. $\hat{\nu}$, denotes the maximum likelihood estimator for the parameter. A double hat, e.g. $\hat{\hat{\nu}}$, denotes the conditional maximum likelihood estimator for the parameter, conditioned on the assumed value of the POI. This test statistic can be used to construct a frequentist confidence interval on the POI. The convention in High Energy Physics (HEP) is to use the CL$_s$ prescription [127, 128], defined as a ratio of $p$-values,

$$ \mathrm{CL}_s = \frac{p_{s+b}}{1 - p_b}, \tag{2} $$

with $p$-values defined as,

$$ p = \int_{t_{\mu,\mathrm{obs}}}^{\infty} f(t_\mu \mid \mu) dt_\mu, \tag{3} $$

---

[5]In general this should correspond to an effective luminosity comparable to the luminosity of any statistically-limited measurements they are to be compared to. For many BSM models the cross sections are small, so this number of events is not enormous. Recent CONTUR publication have for example typically generated the default 30,000 events per set of parameter values.

where $f(t_\mu \mid \mu)$ is the probability density function of the test statistic under an assumed POI. The test statistic in the asymptotic limit [129] can be approximated by

$$t_\mu = \frac{(\mu - \hat{\mu})^2}{\sigma^2} + \mathcal{O}\left(\frac{1}{\sqrt{N}}\right), \tag{4}$$

with $\sigma^2$ the variance of the POI and $N$ the data sample size. Likelihoods in HEP are often written as a Poisson distribution composed of three separate counts; the hypothesised signal count ($s$), the expected background count ($b$) and the observed count ($n$). The POI is now a signal strength parameter. This likelihood is written as,

$$L(\mu) = \frac{(\mu s + b)^n}{n!} e^{-(\mu s + b)}. \tag{5}$$

The form of the test statistic in equation (4) can then be rewritten as

$$\chi_\mu^2 = \frac{((\mu s + b) - n)^2}{\sigma^2} - \frac{((\hat{\mu} s + b) - n)^2}{\sigma^2}, \tag{6}$$

where $\sigma^2$ is now the variance of the counting test. As these are now standard $\chi^2$ distributions, they are approximated by a normal distribution in the large sample limit (via the central limit theorem). This model can be extended by incorporating nuisance parameters on the background model into the likelihood function given in equation (5), and by taking a product of multiple counting tests. A likelihood for a single histogram with $i$ bins and $j$ sources of correlated background nuisance in each bin can be written as,

$$L(\mu, \vec{v}) = \prod_i \frac{(\mu s_i + b_i + \sum_j v_{i,j})^{n_i}}{n_i!} e^{-(\mu s_i + b_i + \sum_j v_{i,j})} \prod_j \exp(\vec{v}_j^\top \Sigma_j^{-1} \vec{v}_j), \tag{7}$$

$$= \prod_i \mathrm{Pois}(\mu s_i + b_i + \sum_j v_{i,j} \mid n_i) \prod_j \mathrm{Gauss}_{iD}(\vec{v}_j \mid 0, \Sigma_j), \tag{8}$$

with $\Sigma_j$ being the covariance matrix for each correlated source of nuisance and $\vec{v}_j$ being the corresponding vector for the correlated nuisance parameters across the bins. In this case there are now multiple sources of nuisance common to each counting test, or bin in the histogram. In this example there would be $j$ different sources of nuisance, so there are $j$ constraint terms. The constraints are now $i$-dimensional Gaussians to account for the covariance of each nuisance parameter between bins. The sources of nuisance can be profiled by maximising the log likelihood for the hypothesised $\mu$.

The practical implementation of this has relied on the inclusion of the uncertainty breakdown into the YODA reference data files included with RIVET. In the asymptotic regime, maximising the log likelihood is equivalent to minimizing the $\chi^2$. The minimisation itself, and handling of covariance information, are achieved with the help of the SCIPY [130] statistics package along with NUMPY [131] for array manipulation. Assuming each individual uncertainty arising from a common named source in the reference data is fully correlated, the correlation matrix for each source of uncertainty can be built. This gives the set of $\Sigma_j$ matrices needed to maximise the likelihood. Minimising the $\chi^2$ for all nuisances simultaneously gives the requisite conditional maximum likelihood estimators, $\hat{\hat{v}}_{i,j}$, required to form the profile likelihood as given in equation (1). Following similar asymptotic arguments, a $\mathrm{CL_s}$ confidence interval can be calculated with the full set of nuisances suitably profiled. As an example, the test statistic in the limiting cases leading to equation (6), for two counting tests with one correlated source of nuisance can be written,

$$\chi_\mu^2 = \begin{pmatrix} \mu s_1 + b_1 + \hat{\hat{v}}_{1,1} - n_1 \\ \mu s_2 + b_2 + \hat{\hat{v}}_{2,1} - n_2 \end{pmatrix}^T \cdot \begin{pmatrix} \Sigma_{11} & \Sigma_{12} \\ \Sigma_{21} & \Sigma_{22} \end{pmatrix}^{-1} \cdot \begin{pmatrix} \mu s_1 + b_1 + \hat{\hat{v}}_{1,1} - n_1 \\ \mu s_2 + b_2 + \hat{\hat{v}}_{2,1} - n_2 \end{pmatrix}. \tag{9}$$

For more complex cases, the sum of the covariance matrices built from each named uncertainty then gives the full covariance matrix between bins which can be used to calculate the likelihood combining all bins in a histogram. Noting that after using the breakdown of the total uncertainty into its component sources to profile the nuisances, the resulting total covariance ($\Sigma = \sum \Sigma_j$) between bins can be used to construct the test. This test statistic omits the second "reference" $\hat{\mu}$ term seen in equation (6), this term is trivial when running the default mode of generating background models from data, however does need full treatment in a similar manner when extending to non trivial background models (see Section 5.3 for more detailed discussion of background models).

The `--correlations` flag can be set in the `contur` executable to enable the calculation to use the correlation information where it is available. The default behaviour of CONTUR is however not to build the correlations between counting tests, and hence fall back to collecting single bins with the largest individual $\mathrm{CL_s}$ to represent the histogram. This is because the full use of correlations can make the main CONTUR run over a large parameter grid quite slow, due to the nuisance parameter minimisation step. Various command-line options are provided to speed up convergence of the fit, or to apply a minimum threshold on the size of considered to error sources, all of which may speed up the process without significantly affecting the result. Neglecting the systematic uncertainty correlations entirely, and falling back on the "single bin" approach for all histograms, is very fast and in most cases gives a result reasonably close to the full exclusion, albeit with more vulnerability to binning effects. The user is encouraged to experiment with these settings, perhaps neglecting correlations for initial scoping scans, and reserving the full correlation treatment for final results.

Currently there is no functionality to correlate named systematics between histograms, which might in principle allow combination of all bins in an entire analysis for example. For most purposes, correlating a given histogram gives the required information. Combining different histograms is then taking a product of the likelihood in equation (7), where the histograms chosen to be combined are deemed to be sufficiently statistically uncorrelated. How these likelihood blocks are chosen to "safely" minimize correlations when combining histograms is described in the following section.

## 5.2 Building a full likelihood

In Section 3.1 the division of available RIVET analyses into pools was shown. As described in Section 3, the data and simulation used for comparison come in the form YODA objects from the relevant HEPDATA entries. Some of these carry information about the correlations between systematic uncertainties. A likelihood of the form given in equation (7) can be used to calculate a representative $\mathrm{CL_s}$ for each histogram.

There are also overlaps between event samples used in many different measurements, which lead to non-trivial correlations in the statistical uncertainties. To avoid spuriously high exclusion rates due to multiple counting of a single unusual feature against several datasets, an algorithm is used to combine histograms safely. To represent this, a pseudo-code realisation of the three main components of the algorithm is given in Listing 5. Starting with an imagined function, `Likelihood`, which would build a likelihood function of a form similar to that described in Section 5.1 from input histogram(s), and return a computed $\mathrm{CL_s}$ value. The stages to combine all the available information into a full likelihood are realised as follows:

1. Calling `BuildFullLikelihood` loops through the defined pools in CONTUR, and calls `EvaluatePool` on each pool.

2. Within each pool, work through all histograms calling `EvaluateHistogram` on each.

3. Depending on desired behaviour `EvaluateHistogram` either builds the correlation

Listing 5: Pseudo-code implementation of the three components of the pool sorting algorithm

```
function BuildFullLikelihood()
      for Pool in ConturPools
            tests append EvaluatePool(Pool)
      Concatenate tests
      return Likelihood(tests)

function EvaluatePool(Pool)
      for Histogram in Pool
            scores append EvaluateHistogram(Histogram)
      Concatenate all orthogonal Histograms
      return Histogram with max(scores)

function EvaluateHistogram(Histogram)
      if Histogram has correlation and (build correlation == True)
            return Likelihood(Histogram)
      else:
            return max(Likelihood(Histogram.bins))
```

matrix where possible, returning the CL$_s$ of the full correlated histogram or defaults to finding the bin within the histogram with the maximum discrepancy, returning this to `EvaluatePool`.

4. Now with each histogram evaluated, `Concatenate` can be called, combining orthogonal counting tests within the pool where allowed. Where a single bin has been used (if no correlation information is requested and or found), the histogram is reduced to this single bin representation. The histogram (or concatenated histogram) with the largest CL$_s$ within the pool is returned to `BuildFullLikelihood`

5. Now the representative histogram from each pool has been appended to a list, this list can also be concatenated. The bins (or bin) extracted from each pool are treated as uncorrelated counting tests with a block diagonal correlation matrix between each pool. The representative CL$_s$ forming the full likelihood can then be returned.

While selecting the most significant deviation within each pool sounds intuitively suspect, in this case it is a conservative approach. Operating in the context of limit setting means that discarding the less-significant deviations simply reduces sensitivity.

## 5.3 Theory-driven background models

The formal argument for a test statistic based on the profile likelihood ratio was given in Section 5.1. An alternative to this would be to use a test statistic based on a simple hypothesis likelihood ratio between the signal ($\mu = 1$) and no signal ($\mu = 0$) hypotheses. Such a test statistic could be written, in a similar form to equation (6), as,

$$\chi'^2_\mu = \frac{((\mu s + b) - n)^2}{\sigma^2} - \frac{(b - n)^2}{\sigma^2},\tag{10}$$

where the background model, $b$, can have nuisance parameters included in a similar fashion which can in turn also be profiled. In the case that the modelled background value, $b$, approaches the observed count, $n$, then the two forms of the test statistic converge. This is equivalent to the statement that as the most likely signal strength ($\hat{\mu}$) tends to zero, the 'reference' values in

the $\chi^2$ test statistics both tend to zero. In this limiting case, the argument followed that the form of the test statistic omitting these reference values, given in equation (9), was sufficient. The example RIVET plot shown in Figure 3 illustrates a signal model appearing in a region of a generated histogram that would however represent different $CL_s$ intervals arising from the two constructions. If the resonance in this spectrum were to appear in one of the regions where the theoretical expectation closely matches the data instead, the two forms would largely coincide.

The default mode of running CONTUR is to generate the background model from the data, and with this the coincidence of the two forms of the test statistic is guaranteed. Typically, state of the art theoretical predictions are not automatically provided alongside measurement data. If such data are provided (as detailed in Section 3.2), then invoking the `--theory` command line option in the `contur` executable will load and use this data where appropriate.

As extensive use of theoretically-generated background models has not yet been made in any physics studies, the default implemented behaviour is to report the $CL_s$ resulting from a direct hypothesis test, essentially as written in equation (9). When more use is made of theoretically-generated "non-trivial" background models in physics studies, it is intended to report both forms of the test statistic as standard. In cases where the nontrivial background model is known to poorly model the data, such as in Figure 3, it is expected that the two forms of the test statistic would start to significantly diverge. The combination of a full profile likelihood with correlated nuisances will enable sophisticated physics studies, however it is expected that the current standard based on a simple 'direct' hypothesis test will remain useful for a range of fast pragmatic studies.

### 5.4 Running the CONTUR likelihood analysis

The method described thus far in this section is handled automatically in the `contur` executable. Either a single YODA file, or a directory containing a structured grid steered by the parameter sampler (as described in Section 4.4), can be supplied to this executable with the `--grid` command-line argument.

In the former case, a summary file is written which may then be processed by the `contur-mk html` script to produce a web page summarising the exclusion and display all the tested RIVET histograms, highlighting those which actually contributed to the likelihood.

In the latter case, the grid will be processed point-wise, evaluating the full likelihood at each parameter point that has been sampled. The resultant grid of evaluated likelihoods is written into a `.map` file, which is a file containing a serialised instance of the `Depot` class. This is written out using the standard library PICKLE functionality and can be read and manipulated for further processing. The PYDOC documentation describing the details of this class is linked from the main CONTUR repository. The executable implements a number of high level control options for vetoing analyses and controlling the statistical treatment.

## 6 Visualisation of parameter space

The `.map` files described in the previous section contain the CONTUR likelihood analysis for a sampled collection of points. The core plotting tools that interact with these files are described in this section. There are multiple auxiliary tools to aid visual understanding of the `.map` files, which are detailed in Appendix C. The core plotting library, which is build upon MATPLOTLIB [134, 135], is implemented in the `plot` module, and user interaction with this module is driven by the `contur-plot` executable. This executable requires three arguments; the `.map` file generated from the main `contur` executable and the names of two parameters on which to draw the axes. Visualisation is limited to two dimensions, but if more than two dimensions were scanned, then multiple 2D plotting instances can be invoked. The names of the requested parameters should

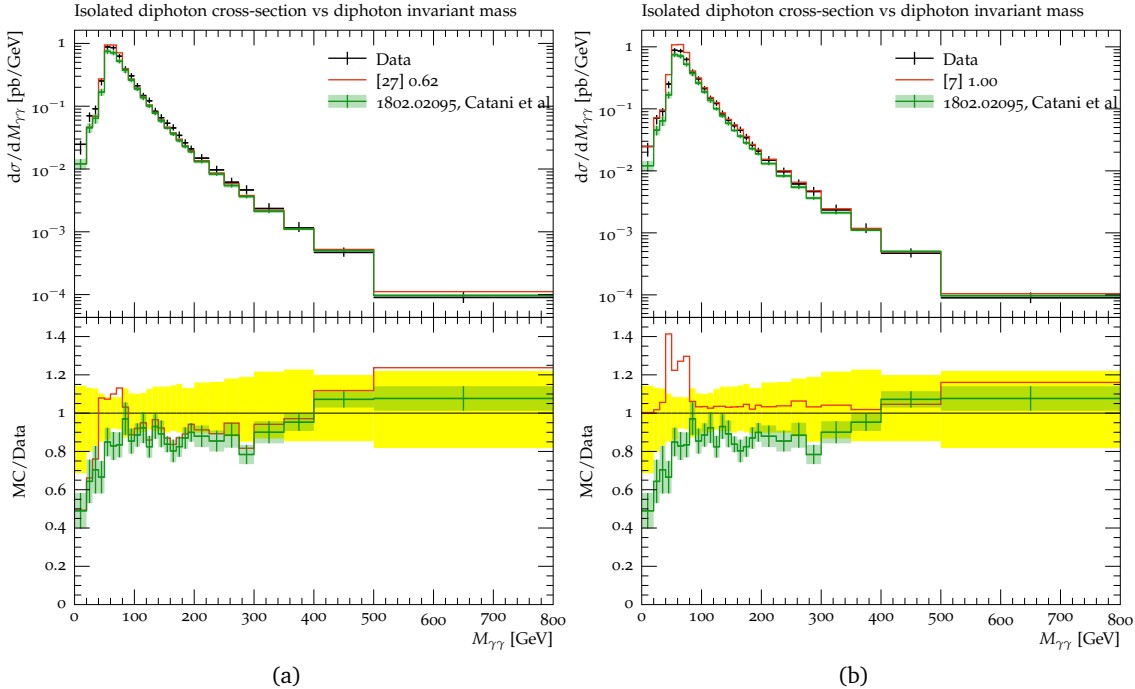

Figure 3: A comparison of a generic Axion-like Particle model [132] producing a 50 GeV resonance in a diphoton mass measurement [87]. The black data points represent the observed data from the original measurement. An NNLO QCD diphoton background prediction is shown in green [133] for comparison. The red line represents the sum of the BSM signal: either with the theoretically calculated background model (Figure 3a), or the background model generated from data (Figure 3b). The caption of the legend for the red line shows the index of the most sensitive bin in square brackets followed by the calculated $CL_s$ value in each case.

match what they were initially called in the parameter sampler (see Section 4.2). The main default visualisation of the likelihood space is demonstrated in Section 6.1. Some methods to interface additional information (such as exclusion contours from other tools) into the default visualisation tools are reviewed in Section 6.2.

## 6.1 Grid visualisation

The sensitivities calculated by CONTUR for each grid point can be expressed as 2D heatmaps, for the overall sensitivity or for each pool separately. The heatmaps indicate where the considered signal model can be excluded due to existing LHC measurements available in RIVET and which part of the phase space is still open. The per-pool heatmaps give more detailed insights into where a specific pool contributes, allowing to draw conclusions on the production processes and decay modes involved. An overview of how the individual pools' sensitivities compare to each other is provided by plotting the dominant pools: CONTUR then shows colour-coded which pool has the highest sensitivity for a given grid point in the same plane as for the heatmaps. Finally, CONTUR also provides exclusion contours at 68% and 95% confidence level as interpolated from the 2D sensitivity grids. Examples of these types of output plots can be found in Figure 4. Further information about available options can be found in Appendix C.7.

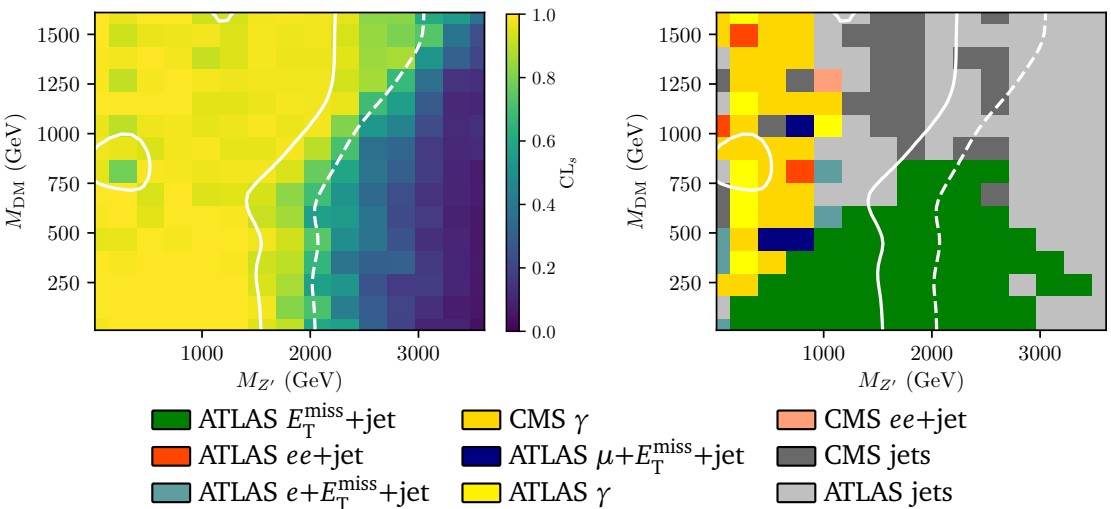

Figure 4: CONTUR 2D heatmaps for a Dark Matter vector mediator model, in the vector mediator mass versus dark matter mass plane. The 95% CL (solid white line) and 68% CL (dashed white line) exclusion limits are superimposed. The plot on the right shows the breakdown into the most sensitive analysis pool for each scan point.

## 6.2 Including additional information in default plots

The default grid visualisation tools described in Section 6.1 provide two methods to supply additional data, allowing creation of additional grids to overlay on the native CONTUR grid. Both methods use the Python IMPORTLIB package, defining a series of Python functions to import via command-line arguments to the `contur-plot` executable. Both methods require user-defined functions that take as an argument a Python dictionary of the parameters, named as specified in the scan (see Section 4). Both methods expect to return a pseudo "exclusion" value specifying the exclusion at the requested point in parameter space. The values are expected to be set such that negative numbers are allowed and positive numbers are excluded (*i.e.* the contour is drawn at the level set of zero), it is generally advised to use the "distance" of the value from 0 to accurately fit the contour. The two methods, and examples of functions expected for the two formats, are given in the following subsections.

### 6.2.1 Plotting external grids

The first method of adding additional information to a plot is invoked by supplying a file containing an external grid with the command-line argument `-eg` (or `--externalGrid`) `NameOfFile`. These functions define user supplied grids, allowing arbitrary numbers of points within the space to be considered. These can be read from additional data sources within the supplied file or simply used to calculate analytic constraints at a much higher resolution than in the CONTUR scan. The function should return a tuple with the first argument being the list of parameter space dictionaries of the new considered points, and the second argument being a list of floats of the pseudo-exclusion values. An example loading in a user supplied grid for visualisation is shown in Listing 6.

### 6.2.2 Plotting external functions

The second method of adding additional information to a plot is invoked by supplying a file containing a function with the command line argument `-ef` (or `--externalFunction`)

Listing 6: An example user-defined grid created in place. The values input would typically be loaded from external files.

```python
def ExampleExternalGrid(paramDict):
    import numpy as np
    pts=[]
    vals=[]

    p1_axes = np.linspace(10,1000.,1)
    p2_axes = np.linspace(10,100.,1)

    for p1 in p1_axes:
        for p2 in p2_axes:
            temp=dict.fromkeys(paramDict)
            temp["contur_p1_name"]=p1
            temp["contur_p2_name"]=p2
            pts.append(temp)
            vals.append(p1/p2-0.5)
    return pts, vals
```

Listing 7: An example user-defined function evaluated on the CONTUR grid points.

```python
def ExampleFunction(paramDict):
    p1=float(paramDict["contur_p1_name"])
    p2=float(paramDict["contur_p2_name"])
return p1/p2-0.5
```

`NameOfFile`. This type of input is for curves which can be evaluated internally from the parameter point coordinates. For example, one might use this functionality to indicate the value of an additional model parameter. These functions recycle the existing CONTUR sample of points to evaluate functions on a grid of the same resolution as the CONTUR scan. The function should return a float of the pseudo-exclusion value. Internally, CONTUR will then evaluate this function on the CONTUR grid. An example function is shown in Listing 7.

## 7 Summary

This manual accompanies the release of CONTUR v2, which is the first public-facing version. Please refer to the CONTUR homepage [10] for links to the latest instructions and source code. In this document, the method and the structure of the CONTUR package were set out, the core functionality of the CONTUR code was described and the motivations behind key design choices were given. On a more philosophical note, the objective of the CONTUR package is to allow the HEP community easily to re-use the LHC analyses preserved in RIVET and HEPDATA to derive exclusions on new physics models. These analyses, the bulk of which are particle-level measurements of SM processes, are often highly model-independent, and can be used to rule out models which would have interfered with otherwise well-understood spectra. The fact that models can be tested programmatically, making use of the runnable code snippets in RIVET which encapsulate their fiducial regions, implies that large regions of parameter space can be probed with minimal "hands-on" effort from analysts. The authors believe that this ability to interrogate existing LHC data directly, rather than construct a new search for each new model proposed by theorists, is a key step in the necessary paradigm shift in HEP from "top-down"

Listing 8: Initiating a docker session

```
$ docker pull hepstore/contur-herwig:latest
$ docker run -it hepstore/contur-herwig
$ [container] setupContur.sh # the user is now within the container
```

(theory-driven) to "bottom-up" (data-driven), which is being brought about by the proliferation of candidate new physics models, in the face of increasingly large datasets and corresponding pressure on computing and human resources. The CONTUR developers are always happy to receive feature requests, and new members of the team are welcome to contribute.

## Acknowledgements

**Thanks**    The authors would like to thank David Grellscheid, Michael Krämer and Björn Sarrazin for helping to realise the first CONTUR proof of principle. We would also like to thank all the colleagues (students, post-docs and academics) who've used development versions of CONTUR and helped us to develop and validate the code in the process.

**Funding information**    Funding sources: MMA, AB, JMB, DY and DH from European Union's Horizon 2020 research and innovation programme as part of the Marie Skłodowska-Curie Innovative Training Network MCnetITN3 (grant agreement no. 722104). AB, JMB, LC and BW from UKRI STFC consolidated grants for experimental particle physics at UCL and Glasgow, and DY from an STFC studentship. AB from Royal Society University Research Fellowship scheme, grant UF160548.

## A    Detailed schematic of CONTUR workflow

A detailed diagram summarising the CONTUR workflow is provided in Figure 5.

## B    Example CONTUR study with HERWIG

CONTUR supports various event generators, as documented in Appendix G, but the default choice is HERWIG. This is because HERWIG features an inclusive event generation mode, where one can very easily generate all $2 \rightarrow 2$ processes which include a BSM particle or coupling. This section shows a complete example of running a parameter scan for a BSM dark matter vector mediator model, encapsulated in the pre-loaded DM_vector_mediator UFO file.

### B.1    Use of DOCKER

As documented in Appendix C.1, the user may find it convenient to run CONTUR within a DOCKER container on a local machine. While this avoids a formal installation of CONTUR's dependencies, it does also prohibit the user from submitting jobs to a HPC cluster: for this, one needs to do a full installation on the relevant cluster. Nonetheless, one can still generate events, run CONTUR on individual parameter points, and analyse results of CONTUR scans which have been performed elsewhere. If the user wishes to use a DOCKER container to run this example, they can follow the commands in Listing 8 and proceed with the rest of the example.



Figure 5: A detailed schematic of the CONTUR workflow. The white boxes indicate steps, including in brackets whether a CONTUR executable, external tool/database, or the user is responsible for execution. The blue boxes indicate intermediary files, including the format. The workflow where a single point of parameter space is analysed is highlighted in the pale orange box, while the workflow to analyse an array of parameter points is highlighted in pale blue. In both cases, the observed data and/or SM background information (boxed in red) are extracted from HEPDATA.

Listing 9: Compiling a UFO file with HERWIG

```
mkdir runarea
cd runarea
cp -r $CONTUR_ROOT/data/share RunInfo
cd RunInfo
cp -r $CONTUR_ROOT/data/Models/DM/DM_vector_mediator_UFO .
ufo2herwig DM_vector_mediator_UFO/
make
```

Listing 10: An example of how to set up a configuration file for HERWIG

```
set /Herwig/FRModel/Particles/Y1:NominalMass 120
set /Herwig/FRModel/Particles/Xm:NominalMass 130
set /Herwig/FRModel/FRModel:gYq 0.1
set /Herwig/FRModel/FRModel:gYXm 0.3
...
insert HPConstructor:Outgoing 0 /Herwig/FRModel/Particles/Y1
insert ResConstructor:Intermediates 0 /Herwig/FRModel/Particles/Y1
set HPConstructor:Processes SingleParticleInclusive
...
set EventGenerator:EventHandler:LuminosityFunction:Energy 13000
create ThePEG::RivetAnalysis Rivet RivetAnalysis.so
insert EventGenerator:AnalysisHandlers 0 Rivet
read 13TeV.ana
saverun LHC EventGenerator
```

## B.2 Setting up the run area

The model used in this example is one of many pre-loaded example UFO files and associated templates that come with the CONTUR package. These can be found in the contur/data/Models directory. The first step is to make a work area, and to copy into it a template RunInfo directory as well as the model's UFO files. Once this has been done, one needs to convert the UFO to the HERWIG format, and compile it. This will render the model readable by HERWIG. Listing 9 shows the steps to setup a run area for a DM vector mediator model.

## B.3 Event generation

In HERWIG, an EventGenerator object is built to generate events. The configuration for this object is done in a HERWIG input file (see Listing 10 for an example), with filename extension .in. In CONTUR these files are usually called LHC.in, and for each example model CONTUR provides there is an associated example input file in the model directory. A recommended starting point for CONTUR studies is to complete a single run of HERWIG on the chosen model.

The LHC.in file needs to be customized for a particular model, by specifying the values of the parameters in the UFO file one is considering. This might mean setting particle masses, coupling strengths or other model parameters. The input file should therefore contain lines like those in Listing 10, customized to the parameters of a given model. One should also specify which BSM particles should be considered during event generation (either as outgoing or intermediate particles), and add the setting to inclusively generate all processes involving that particle. Finally, one needs to tell HERWIG to pipe the generated events into RIVET, so that they can be analysed directly and used in the CONTUR workflow. For batch runs, CONTUR's steering code automatically appends these lines.

Listing 11: Generating events with HERWIG

```
cd runarea
cp RunInfo/DM_vector_mediator_UFO/LHC-test.in LHC.in
Herwig read LHC.in -I RunInfo -L RunInfo
Herwig run LHC.run -N 200
```

Listing 12: Running CONTUR on a single YODA file

```
cd runarea
contur LHC.yoda
contur-mkhtml LHC.yoda
```

After setting up the input file, one can simply read it and generate events with HERWIG. For the case of the DM vector mediator model, a template HERWIG configuration file for a single run is provided in the model directory. Listing 11 shows the steps to using this template for event generation. First one must copy the template file into the run area. Next, the HERWIG read step reads and builds the event generator from the configuration file LHC.in, and the HERWIG run line tells the event generator to generate 200 events. Note that the HERWIG run card, LHC.run, is the output of the first line. If successfully run, the output of Listing 11 will produce the file LHC.yoda containing the results of the HERWIG run. Note that the commands here will read the analysis listing file 13TeV.ana from the installed area. If you wish instead to read a local version, modify the read command -I argument to point to that instead.

### B.4 Running CONTUR on a single YODA file

Following the steps of the previous section where an LHC.yoda file was produced from the DM vector mediator model, the second command in Listing 12 tells CONTUR to analyse it. The computed exclusion will be printed to the terminal. Additional options when running CONTUR are also available, and can be accessed through contur --help.

Once CONTUR has successfully analysed the YODA file, an ANALYSIS folder is made, and an exclusion for the model at the specified parameter points is printed alongside some other information about the run. Following along the case study for the DM vector mediator model, the printed exclusion corresponds to the parameter values defined in Listing 10. It is often useful to plot the relevant RIVET histograms from the CONTUR run to get a better idea of the underlying physics in the calculated exclusion. The third line of Listing 12 runs contur-mkhtml on the analyzed YODA file, which generates a contur-plots directory that contains all the histograms, alongside an index.html file to view them in a browser. An example of the output is shown in Figure 6.

### B.5 Setting up for CONTUR batch jobs (HPC system)

*This step cannot be run from within a DOCKER container, as it requires access to a HPC system. Instead, one should use a full installation on a HPC cluster.*

The next step is to essentially repeat the procedures described in Appendix B.3 and Appendix B.4 and complete a series of runs at various model parameter points, so that an exclusion for the model in the parameter space can be drawn as a 2D heatmap. This can be done efficiently using CONTUR's automated batch-job submission functionality. Here we assume that qsub is available on your system; it is the default choice for CONTUR. Slurm and HTCondor batch

Listing 13: Setting up to run a CONTUR batch job

```
cd runarea
cp RunInfo/DM_vector_mediator_UFO/LHC-example.in LHC.in
cp RunInfo/DM_vector_mediator_UFO/param_file.dat .
```

Listing 14: An example of how to set up a configuration file for HERWIG batch runs

```
    set /Herwig/FRModel/Particles/Y1:NominalMass {mY1}
    set /Herwig/FRModel/Particles/Xm:NominalMass {mXm}
    set /Herwig/FRModel/FRModel:gYXm {gYXm}
    set /Herwig/FRModel/FRModel:gYq {gYq}
    ...
    insert HPConstructor:Outgoing 0 /Herwig/FRModel/Particles/Y1
    insert HPConstructor:Outgoing 0 /Herwig/FRModel/Particles/Xm
    set HPConstructor:Processes SingleParticleInclusive
```

systems are also supported, as described in Section 4.4.

To set up CONTUR for batch-job submission, the user must tell CONTUR what region of parameter space to sample. To do this, the user should replace the nominal mass of dark matter (Xm), nominal mass of the vector mediator (Y1), and the couplings gYq and gYXm in Listing 10 by arbitrary variables. A steering file, `param_file.dat`, will specify what values to set for each parameter. Template files are available in the `data/share` directory and can be copied to `runarea` as per Listing 13.

Listing 15: CONTUR steering file for DM vector mediator model

```
    [Run]
    generator ='/path/to/generatorSetup.sh'
    contur ='/path/to/setupContur.sh'

    [Parameters]
    [[mXm]]
    mode = LIN
    start = 10.0
    stop = 1610.0
    number = 10
    [[mY1]]
    mode = LIN
    start = 10.0
    stop = 3610.0
    number = 10
    [[gYXm]]
    mode = CONST
    value = 1.0
    [[gYq]]
    mode = CONST
    value = 0.25
```

The newly-copied LHC.in should resemble Listing 14, where the values for the model parameters have been replaced with their respective variables inside curly brackets. Also notice that this version of the HERWIG input file is missing the commands that specify beam energy, run the RIVET pipeline, and save the event generator (cf. listing 10). These lines will be added automatically by CONTUR for each beam energy when the batch-jobs are submitted.

Listing 16: Submitting a CONTUR batch job with 1000 events per point to the `mediumc7` batch queue (queue names will of course depend on your local cluster)

```
cd runarea
contur-batch -n 1000 -q mediumc7
```

Listing 17: Running CONTUR on a grid

```
cd runarea
contur -g myscan00
```

The user should then modify the steering file `param_file.dat` to look like listing Listing 15, replacing the placeholder paths for HERWIG and CONTUR under the `Run` heading with local paths. The free parameters of our DM vector mediator model are listed under `Parameters`. The variable name for each parameter must match those in Listing 14 for CONTUR to recognize the parameter and substitute in the correct value.

For each parameter, the mode for sampling must be specified. For this example, the particle masses of the dark matter candidate and the vector mediator are set to the `LIN` linear mode. The start and stop options indicate the sampling range in GeV, and `number=15` tells CONTUR to sample 10 points in this range in a linear fashion. The coupling of the vector mediator to dark matter and to quarks are set to the `CONST` constant mode, with values 1.0 and 0.25 respectively.

The user is now ready to submit a batch job over the specified range of parameter points, following the commands of Listing 16 in the `runarea` directory. This will create a directory called `myscan00` which contains three directories corresponding to the beam energies 7, 8, and 13 TeV. Inside each beam energy directory there will be 100 run-point subdirectories that correspond to your specified range in `param_file.dat`. Inspecting a few of these reveals the HERWIG input files generated, and the shell scripts (`runpoint_xxxx.sh`). These shell scripts can be submitted manually, or run locally in your terminal for troubleshooting.

### B.6 Running CONTUR on a grid

Once the batch jobs have successfully finished, each runpoint directory in `myscan00` should have produced a YODA file. Each YODA file will be named according to the scan point, for example `LHC-S101-runpoint_0001.yoda`. Some functionalities are provided by `contur-gridtool`, which performs various manipulations on a completed CONTUR grid and can be useful for troubleshooting. For example, running `contur-gridtool -c myscan00` runs a check to see whether all grid points have valid YODA files. See section C.3 for full details of how to use it.

Listing 17 runs CONTUR with the `-g` or `--grid` option, which means statistical analysis is to be performed on the specified grid `myscan00`. The output of this step is a directory named `ANALYSIS`, inside which a `contur.map` file of the corresponding grid is produced. You may wish to create a shell script for this step and submit it to the batch system for larger grids.

### B.7 Making CONTUR heatmaps

The last step in a CONTUR sensitivity study is the visualisation of the computed limits in the form of 2D sensitivity heatmaps. Once CONTUR has successfully run and produced a `contur.map` file, one can run the `contur-plot` command on it while specifying the variables to plot. In the example in Listing 18, the mass of the dark matter particle `mXm` is on the $x$-axis and the mass of the vector mediator `mY1` is on the $y$-axis. The output plots are shown in Figure 4.

<p align="center">Listing 18: Plotting 2D heatmaps with CONTUR</p>

```
cd runarea/ANALYSIS
contur-plot contur.map mXm mY1
```

## C  CONTUR tools and utilities

The CONTUR package provides several tools and executables to assist the user in the preparation, manipulation and visualisation of CONTUR results. These tools are documented below.

### C.1  CONTUR DOCKER containers

Containerisation of software packages with DOCKER provides a convenient way to bundle a piece of software with all its dependencies, so that one can forego a formal installation and run the software simply by downloading and entering the container. This also allows fast and trouble-free deployment across operating systems. The CONTUR developers maintain two types of container, which are regularly updated on DOCKERHUB:

- `hepstore/contur` is a container which includes the latest version of CONTUR along with all its dependencies *except* for the HERWIG MCEG. This is useful for users who do not wish to generate events, but instead analyse the results of existing scans performed elsewhere, or make use of the visualisation tools. It may also be of use for expert users who wish to use the container as a base and install other MCEGs on top.

- `hepstore/contur-herwig` is a container which includes the latest version of CONTUR along with all its dependencies *including* the HERWIG MCEG. This is heavier than the `hepstore/contur` container, but allows the user to generate events.

These containers can be downloaded from the command line using for example `docker pull hepstore/contur-herwig:latest`, where the tag after the colon can be replaced by another keyword to download a particular version.

A limitation of running CONTUR via a DOCKER container is that one does not typically have access to a HPC on a local machine. Furthermore, HPC clusters often do not support jobs running through a DOCKER container. Therefore, it is usually not possible to submit scans to HPC clusters if using CONTUR via a container.

If the user wishes to build their own container locally, they can make use of the `Dockerfiles` which are provided in `docker/*/Dockerfile`. In addition to the `Dockerfiles` used for the above-listed containers, a `contur-dev` Dockerfile is provided, for users wishing to access the development branch of CONTUR. A detailed example of how to download or build one of the CONTUR containers is provided in Listing 19.

### C.2  Exporting the results of a CONTUR statistical analysis to a CSV file using `contur-export`

The map files containing the CONTUR likelihood analysis for a sampled collection of points described in Section 5.4 can also be exported to a CSV file by using the `contur-export` command with the `-i` and `-o` flags to specify the input and output paths for the map and CSV files respectively. Adding the `-dp` or `--dominant-pools` flag appends a column containing the dominant pools for each point.

Listing 19: Examples of how to download and run CONTUR via a DOCKER container assuming DOCKER is installed on the machine (instructions to install DOCKER on a variety of operating systems are available online).

```
$ docker pull hepstore/contur-herwig:latest
# Or: docker pull hepstore/contur:latest for the version with Herwig.
# One can also substitute the "latest" tag for others corresponding to particular versions.

$ docker run -it -p 80:80 -v path/to/useful/directory:/mydir contur-herwig
# The −it flag indicates interactive mode: the user enters a shell within the container and runs
# the software from there.
# −p 80:80 exposes port 80, for use if the user wishes to make use of the interactive visualisation
# tool contur−visualiser.
# The −v option maps some directory on the local machine to a directory within the container,
# so that files can be accessed or written out. This feature may be used to import UFO files, write out
# plots and map files, or even to develop the Contur code from within the container.
$ [container] source setupContur.sh  # note the user shell is now within the container
$ [container] ...  # proceed with desired studies
$ [container] exit  # exit when done

# One may instead wish to build their own container locally:
$ git clone https://gitlab.com/hepcedar/contur.git
$ cd docker/contur  # or e.g. docker/contur−dev, to directory containing Dockerfile
$ docker build -t contur .  # it may then take some time to build the container...
$ docker run -it -p 80:80 -v path/to/useful/directory:/mydir contur
# ... and proceed as above
```

### C.3   Manipulating CONTUR scan directories with `contur-gridtool`

CONTUR provides a compilation of grid tools for managing grids produced with the `contur-batch` command. These can be accessed with the `contur-gridtool` command followed by various optional parameters. They allow to merge different signal grids into a single one (`--merge`), removing files unnecessary for post-processing (`--remove-merged`) or compressing those that are crucial to reduce disk space (`--archive`). Other options check for (`--check`) or resubmit (`--submit`) failed jobs to the batch system or identify the grid points that are most consistent with a given set of parameters (`--findPoint`).

### C.4   Concatenating the results of CONTUR statistical analyses

Running the `contur` command on a scan directory produces a `.map` file. One may want to concatenate the results of several scans by merging their relevant `.map` files. This can be achieved using the `contur-mapmerge` command.

### C.5   Submitting CONTUR scans to a HPC systems using `contur-batch`

An executable called `contur-batch` is provided to prepare a parameter space scan and submit batch jobs. It produces a directory for each of the various beam energies (7, 8 and 13 TeV by default, but configurable with the `--beams` command), containing generator configuration files detailing the parameters used at that run point and a shell script to run the generator that is then submitted to a HPC cluster. The `--param_file`, `--template_file` and `--grid` options may be used to specify the names of the relevant configuration files if they differ from the defaults. The number of events to generate at each point is controlled by the `--numevents` option, defaulting to 30, 000. In the simplest use-case, the same number of events will be generated at

each point in the scan. However, this may be sub-optimal, since some areas of parameter space may require far more events than others, for example if BSM processes are swamped by SM processes. In such a case, the expert user may wish to generate a different number of events at each point. This behaviour can be enabled using the `--variablePrecision` flag, which then looks for an additional section of the parameter card entitled `NEventScalings` indicating how to scale the number of events for each point. See Section 4.2 for more information on the parameter card, and Section C.6 for information on the `contur-zoom` tool which can be used to automatically add the `NEventScalings` section to a parameter card.

By default, the tool assumes that HERWIG is the MCEG, but this can be changed with `--mceg`. If the user wishes to run their own instances of a MCEG and RIVET, and pipe this information to the jobs, the flag `--pipe-hepmc` can be used. The MCEG seed can be changed using the `--seed` option.

One can use the `--out_dir` option to specify where the scan directory should be written. Several options exist to specify the batch system to use (`--batch_system`) and the queue name (`--queue`) and well as the maximum wall time (`--walltime`) and maximum memory consumption for jobs (`--memory`). Finally, the `--scan-only` flag can be used to do a dry run: prepare the directories without submitting them to the cluster.

## C.6 Iteratively refining the scanned parameter space with `contur-zoom`

The `contur-zoom` utility is designed to optimise the hyper-parameters of a parameter scan, such as the ranges, granularity of binning, and number of events to generate at each point. The reasoning behind this tool is that not all areas of a parameter space are interesting: indeed, parts of the parameter space which are well below the exclusion level, or well above it, can be ignored, and the more interesting regions to focus on are those where the gradient of $CL_s$ values is large. Furthermore, focusing only on interesting regions of parameter space avoids wasting computing resources on points where the result is unambiguous. A user approaching a new model may wish to begin with a coarse, wide-ranging scan of parameter space, and then iteratively "zoom" into the more interesting regions.

`contur-zoom` automatically determines a new set of hyper-parameters for a parameter card, given the results of a previous coarser scan. It does this by defining a figure of merit for each point in a scan, to approximate the $CL_s$ gradient at that point; this is calculated as the average difference in the $CL_s$ value of a given point with respect to all adjacent points in the scan. By construction, this figure of merit will always be within 0 and 1, since that is the range of possible $CL_s$ values. This figure of merit is implemented for CONTUR scans of arbitrarily-high dimensionality.

Given a figure of merit for each point, it is possible to change the ranges of a scan to focus on the region with the fastest change in $CL_s$, by specifying a minimum threshold for the figure of merit. The "zoomed" range of parameter values is an $n$-dimensional parameter space that is obtained by iterating through each dimension, and locating the smallest range of points on that axis that contains all points above the threshold, and using this reduced parameter space for the next iteration. The result is a new rectilinear scan range of the model parameters, containing all points with a figure of merit above the threshold.

This procedure can be applied to a CONTUR parameter scan using the `contur-zoom` command, where the threshold can be specified with `--thresh`, defaulting to 0.25. The `.map` file for the original scan should be provided using `--m_path` or `-m`, and the corresponding original parameter file with `--o_path`. One can either choose to replace the original parameter file with the zoomed version (using `--replace`), or specify where to write the new files to with `--n_path`. If one wanted to restrict the zooming to a single dimension of the $n$-dimensional parameter space, this can be achieved using the `--param` option. Finally, one can over-ride the figure of merit for particular points with special $CL_s$ values, so that they are included in

the new range regardless of the gradient. For example, one may wish to keep all points on the 68% and/or 95% CL contours. This can be achieved using the `--vals` option, and specifying a space-separated list of $CL_s$ values (between 0 and 1). The algorithm will then keep all points with $CL_s$ within 0.01 of the specified values.

To avoid wasting computing resources on uninteresting points, one may consider excluding points with a figure of merit below a given threshold from processing. This can be achieved using the `--skipPoints` option of `contur-zoom`, with the exclusion threshold specified by `--thresh`. The indices of all points below that threshold will be added to a new block of the parameter card, labelled `SkippedPoints`, and these points will not be processed during the CONTUR scan directory preparation and processing.

Finally, one may want to prepare a variable-precision scan over a parameter space, *i.e.* one where the number of events generated at each point may change depending on the region of parameter space. One may wish to generate more events near "interesting" regions, according to the figure of merit defined above. This can be achieved using the `--nEventScalings` option. This option will add a section to the parameter card labelled `NEventScalings`, which is simply the value of the figure of merit at each point. When using `contur-batch` with a parameter card which has a `NEventScalings` section, and using the the `--variablePrecision` (or simply `-v`) option to indicate a variable-precision scan, the number of events specified with `-n` will indicate a maximum number of events, which will be scaled by the value of the figure of merit at that point. Thus, the points with the highest figure of merit (which is always between 0 and 1 by construction) will be processed with a number of events close to the maximum, while less interesting points, where the $CL_s$ gradient is smaller, will be generated with fewer events.

Finally, the `contur-zoom` command allows the user to re-bin the parameter space based on the figure of merit, while maintaining the same ranges. This can be achieved using the `--rebin` flag. This will create a new parameter card where the number of bins along each axis is doubled. This option can be used in tandem with the `--skipPoints` and `--nEventScalings` options, where the figure of merit of the newly-generated bins will be by default set to the same value as their parent bins.

Examples of the effect of `contur-zoom` commands on an example parameter card can be seen in Listing 20. The suggested approach would be to begin with a broad, coarse scan over a given parameter space, and iteratively update the ranges, number of points and number of bins using `contur-zoom`.

## C.7 Visualising the results of a CONTUR scan using `contur-plot`

The `contur-plot` executable produces visualisations of the results of a CONTUR grid scan. The tool takes as input a `.map` file obtained from running `contur -g` on a parameter scan directory. This executable can handle 2- or 3-dimensional scans. The user should therefore specify a `.map` file to read and 2 or 3 variables to plot as positional arguments.

In addition to the positional arguments, the user can specify `--theory` and `--data` arguments to add additional information to a plot. This is discussed in detail in Section 6.2.

The plot title can be set using the `--title` option. The $x$- and/or $y$-axis labels can be set using `--xlabel` and `--ylabel` options, which accept LaTeX formatting but special characters must be escaped with a backslash. Furthermore, the user may choose to display the $x$- and/or $y$-axis on a logarithmic scale using `--xlog` and `--ylog` flags. In addition to the overall heatmaps, the heatmaps for individual analysis pools can be generated if the `--pools` flag is turned on, where certain pools can be skipped using the `--omit` option.

Some other expert-user options exist to control the interpolation between points, for example. The full list can be viewed using `--help`. A few examples of `contur-plot` commands are shown in Listing 21.

Listing 20: Examples of the usage and output of the `contur-zoom` command.

```
$ contur-zoom --thresh 0.3 --o_path param_file.dat --n_path param_file.
    zoomed.dat --m_path ANALYSIS/contur.map
$ cat param_file.zoomed.dat
[Run]
generator = "/path/to/generatorSetup.sh"
contur = "/path/to/setupContur.sh"
[Parameters]
[[x0]]
mode = LIN
start = 300.0
stop = 1000.0
number = 15
[[x1]]
mode = CONST
value = 2.0

$ contur-zoom --thresh 0.05 --skipPoints --o_path param_file.zoomed.dat --
    n_path param_file.zoomed.excl.dat --m_path ANALYSIS/contur.map
$ cat param_file.zoomed.excl.dat
[Run]
generator = "/path/to/generatorSetup.sh"
contur = "/path/to/setupContur.sh"
[Parameters]
[[x0]]
mode = LIN
start = 300.0
stop = 1000.0
number = 15
[[x1]]
mode = CONST
value = 2.0
[SkippedPoints]
points = 5 6 7 21 22 25 26 29 30 33 34 37 38 45 46 47 48 49 50 51 52 #...

$ contur-zoom --nEventScalings --skipPoints --o_path param_file.zoomed.excl.
    dat --n_path param_file.zoomed.excl.scaled.dat --m_path ANALYSIS/contur.
    map
$ cat param_file.zoomed.excl.scaled.dat
[Run]
generator = "/path/to/generatorSetup.sh"
contur = "/path/to/setupContur.sh"
[Parameters]
[[x0]]
mode = LIN
start = 300.0
stop = 1000.0
number = 15
[[x1]]
mode = CONST
value = 2.0
[SkippedPoints]
points = 5 6 7 21 22 25 26 29 30 33 34 37 38 45 46 47 48 49 50 51 52 #...
[NEventScalings]
points = 0.16713080576991923 0.16713080576991923 0.16713080576991923
    0.2614096406084278 0.08944325063426813 ..
```

Listing 21: Examples of the usage of the `contur-plot` command.

```
$ contur-plot myscan.map Xm Y1  # to plot heat map with Xm on x−axis and Y1 on y−axis

$ contur-plot myscan.map Xm Y1 gYq -s 50  # to plot Xm on x−axis, Y1 on y−axis and slice 50
# of gYq (z−axis)

$ contur-plot myscan.map Xm Y1 gYq -sc  # to plot 3d scatter graph

$ contur-plot myscan.map Xm Y1 gYq -a  # to plot Xm on x−axis, Y1 on y−axis and all slices
# of gYq
```

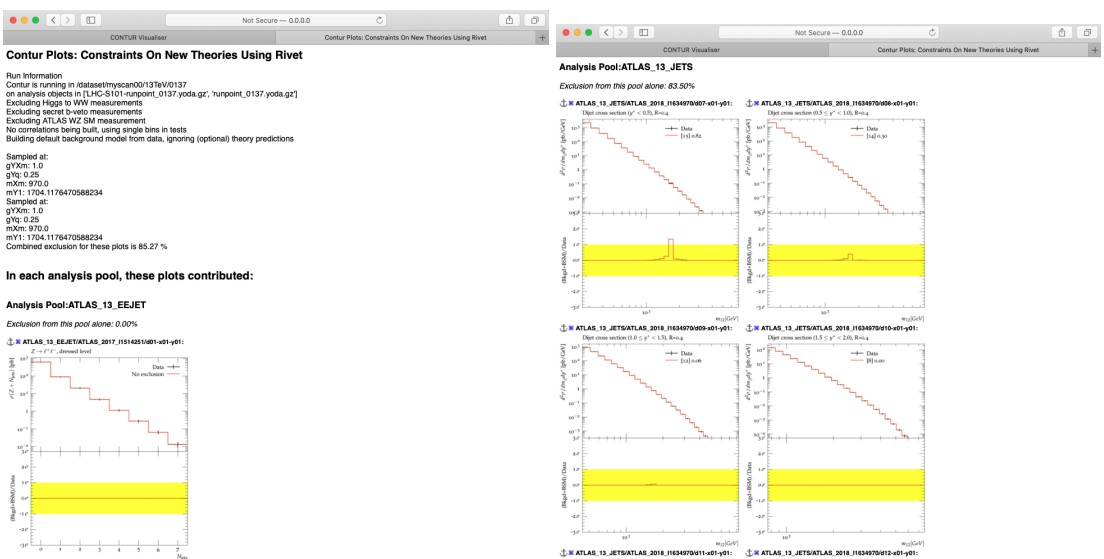

Figure 6: Example of the summary web page for a single-point `contur` run, as produced by `contur-mkhtml`

.

## C.8  Visualising a single parameter point with `contur-mkhtml`

It is often useful to run CONTUR on a single parameter point (*i.e.* a single YODA file) from a scan, to understand which analyses are providing the exclusion power, and view the associated histograms super-imposing the measured data and the generated signal at that point. The `contur-mkhtml` utility prepares a summary page for the user, which concisely presents the most important histograms which contribute to the CL$_s$ exclusion at a given point.

The executable can only be used after the main `contur` executable has been run on the YODA file of a given point, as in the example in Listing 12. An example screenshot of the summary web page which is produced can be seen in Figure 6. The `--reducedPlots` flag causes only the most important histograms to be plotted, thus speeding up processing time.

## C.9  HERWIG-specific cross-section visualisation tools

For each point in a scan of parameters, HERWIG produces log files which detail the generated cross-sections and branching ratios for the processes which contribute. This is valuable information, as one can use it to understand how the phenomenology of the model changes for different regions of the parameter space. Two helper Python executables are provided, to parse this information and present it in a digestible format.

First, `contur-extract-herwig-xs-br` parses the log files for a given point in the param-

eter scan, and returns an ordered list of processes and their cross-sections to the terminal. This list represents all the processes which contribute some configurable fraction of the total cross-section (option `--tolerance`, by default 1%) at that point. To aid digestion of results, similar processes are grouped together, with their cross-sections summed. The default summation rules are summarised below.

- Differences between leptons (electrons, muons and taus) are ignored. This behaviour can be over-ridden using `--splitLeptons, --sl`;

- Differences between incoming partons are ignored: all flavours of quarks and gluons are merged. This behaviour can be over-ridden using `--splitPartons, --sp`;

- Differences between particles and antiparticles are ignored. This behaviour can be over-ridden using `--splitAntiparticles, --sa`;

- Differences between "light" quarks are ignored: $u, d, s, c, b$ are grouped. This behaviour can be over-ridden using `--splitLightQuarks, --sq`, or just the $b$ can be split out using `--split_b, --sb`;

- Optionally, one can choose to ignore differences between electroweak bosons $W, Z, H$ using `--mergeBosons, --mb`;

The resulting output will show the outgoing particles from the matrix element. It may be that one is interested not in the particles which come out of the hard scatter, but the stable particles one would find in the final state. To help determine this information, `contur-extract-herwig-xs-br` can recursively apply SM branching fractions of unstable SM particles, and can extract the predicted branching fractions of BSM particles from the log files and apply those recursively too. This behaviour can be activated using the `--foldBRs,--br` option or `--foldBSMBRs, --br_bsm` for BSM decays only. Some examples of the output of the script can be found in Listing 22.

A second executable, `contur-scan-herwig-xs-br`, can call `contur-extract-herwig-xs-br` at each point in a parameter scan, and present the cross-section information as a cross-section heatmap for each process. At present, the tool can only handle two-dimensional scans, and the variables to use as the $x$- and $y$-axes of the resulting plots should be provided via `--xy`. This script takes the same options as `contur-extract-herwig-xs-br` in terms of merging similar processes, and additionally can take a `-p` or `--pools` option, which further groups final states into pools of "analysis types", for example grouping together processes which have the same or similar number of leptons, photons, jets (from quarks, or gluons) or $b$-jets, or missing energy (from neutrinos or stable BSM particles). Examples of outputs of this tool can be found in Figure 7.

### C.10 Interactive visualisation

To further aid digestion of CONTUR results, a web-based visualisation tool, `contur-visualiser`, is provided. This tools combines the CONTUR parameter scan, $CL_s$ calculation and HERWIG log-file parsing, to build an interactive webpage where these results are presented in a combined way. The page can be opened on any browser on the local machine. The result is a heatmap showing the $CL_s$ exclusion at each point of a parameter scan, where hovering the cursor over a particular point reveals the cross-section information for that point. Clicking on a given point will cause the evaluation of a single-point CONTUR run (as described in Section C.8), and will open a new page showing the summary for that point.

Since the visualiser is I/O intensive, it is recommended to run the `contur-visualiser` locally rather than via `ssh`. Users may find it convenient to run this tool via a DOCKER image (the

Listing 22: Examples of the usage and output of the contur-extract-herwig-xs-br script.

```
[myscan00]$ contur-extract-herwig-xs-br -i 13TeV/0001/ --tolerance 0.0
13TeV/0001 :: gYXm=1.000000, gYq=0.250000, mXm=116.666667, mY1=10.000000
totalXS 167000000.00 fb
145000000.00 fb, (86.83%), p p \rightarrow Y1 q
14500000.00 fb, (8.68%), p p \rightarrow Y1 g
5500000.00 fb, (3.29%), p p \rightarrow Y1 Y1
900000.00 fb, (0.54%), p p \rightarrow W Y1
700000.00 fb, (0.42%), p p \rightarrow q q
300000.00 fb, (0.18%), p p \rightarrow Y1 \gamma
200000.00 fb, (0.12%), p p \rightarrow Y1 Z

[myscan00]$ contur-extract-herwig-xs-br -i 13TeV/0001/ --tolerance 0.0 \
--splitPartons --splitAntiparticles
13TeV/0001 :: gYXm=1.000000, gYq=0.250000, mXm=116.666667, mY1=10.000000
totalXS 167000000.00 fb
105000000.00 fb, (62.87%), g q \rightarrow Y1 q
40000000.00 fb, (23.95%), \bar{q} g \rightarrow Y1 \bar{q}
14500000.00 fb, (8.68%), \bar{q} q \rightarrow Y1 g
5500000.00 fb, (3.29%), \bar{q} q \rightarrow Y1 Y1
900000.00 fb, (0.54%), \bar{q} q \rightarrow W Y1
700000.00 fb, (0.42%), \bar{q} q \rightarrow \bar{q} q
300000.00 fb, (0.18%), \bar{q} q \rightarrow Y1 \gamma
200000.00 fb, (0.12%), \bar{q} q \rightarrow Y1 Z

[myscan00]$ contur-extract-herwig-xs-br -i 13TeV/0001/ --tolerance 0.0 \
--splitPartons --splitLightQuarks --mergeBosons
13TeV/0001 :: gYXm=1.000000, gYq=0.250000, mXm=116.666667, mY1=10.000000
totalXS 167000000.00 fb
83000000.00 fb, (49.70%), g u \rightarrow Y1 u
62000000.00 fb, (37.13%), d g \rightarrow Y1 d
9000000.00 fb, (5.39%), u u \rightarrow Y1 g
5500000.00 fb, (3.29%), d d \rightarrow Y1 g
3700000.00 fb, (2.22%), u u \rightarrow Y1 Y1
1800000.00 fb, (1.08%), d d \rightarrow Y1 Y1
900000.00 fb, (0.54%), d u \rightarrow V Y1
700000.00 fb, (0.42%), d d \rightarrow u u
300000.00 fb, (0.18%), u u \rightarrow Y1 \gamma
200000.00 fb, (0.12%), d d \rightarrow V Y1

[myscan00]$ contur-extract-herwig-xs-br -i 13TeV/0001/ --tolerance 0.0 --
    foldBRs
13TeV/0001 :: gYXm=1.000000, gYq=0.250000, mXm=116.666667, mY1=10.000000
totalXS 167000000.00 fb
145000000.00 fb, (86.83%), p p \rightarrow q q q
14500000.00 fb, (8.68%), p p \rightarrow g q q
6250000.00 fb, (3.74%), p p \rightarrow q q q q
700000.00 fb, (0.42%), p p \rightarrow q q
300000.00 fb, (0.18%), p p \rightarrow \gamma q q
288000.00 fb, (0.17%), p p \rightarrow \nu l q q
41000.00 fb, (0.02%), p p \rightarrow \nu \nu q q
21000.00 fb, (0.01%), p p \rightarrow l l q q
```

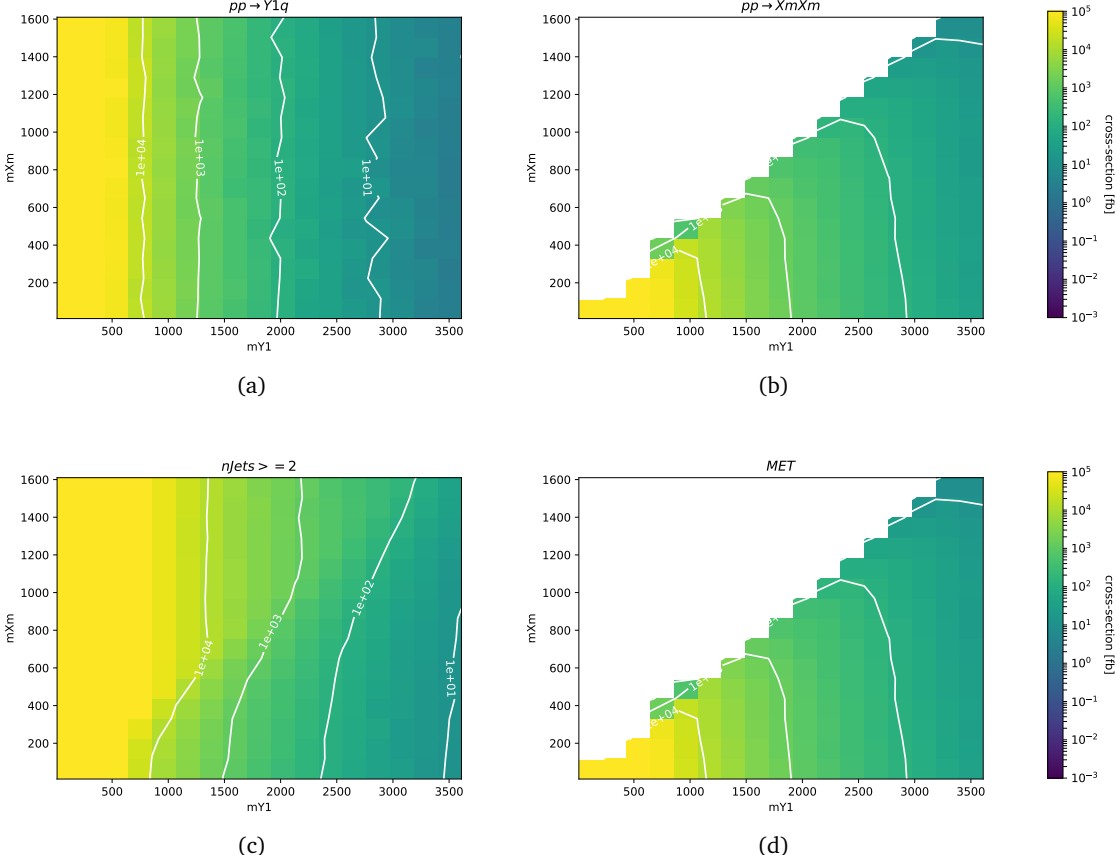

**Figure 7:** Leading-order cross-sections extracted from HERWIG logs using the `contur-scan-herwig-xs-br` tool. The top row corresponds to the default output for the tool, showing the three dominant $2 \rightarrow 2$ processes across the scanned parameter space which contain a BSM particle in an outgoing leg or in the propagator. The bottom row corresponds to the output of the tool with `--br --pools` options activated. The stable BSM particles are assumed to show up as missing transverse energy (MET). These plots reveal that the production of certain particles, and certain decay modes, are only accessible in certain regions of the parameter space, giving an insight of how the phenomenology of the model varies from region to region.

necessary `Dockerfile`s are available in `/contur/docker/contur-visualiser`). When initialising the DOCKER image, one should map the location of the scan files and map (which may have been downloaded from a cluster), as well as exposing port 80 to the DOCKER container via the DOCKER option `-p 80:80`, so its output can be accessed via HTTP.

Once inside the container, the `contur-visualiser` tool should be run from the `contur-visualiser` directory, and requires a path to a `.map` file (using `-m`), the path to the scan directory (using `-d`), and the names of the variables to plot (using `-x` and `-y`).

The visualiser will then run through each point in the parameter scan directory and collect the output of `contur-extract-herwig-xs-br`, as well as the $CL_s$ values at each point. Then, it will create an interactive page which can be accessed by opening `http://0.0.0.0:80/` on the local machine (outside the DOCKER container). An example of a screenshot of such a web page is provided in Figure 8. Hovering over the points on the heatmap reveals the $x$, $y$ and $CL_s$ values at that point, while the side-panel shows the `contur-extract-herwig-xs-br` output at that point, so that the user can gauge which processes might be contributing to

Listing 23: Detailed guide to using the `contur-visualiser` tool.

```
# One first needs to build the Docker container with the correct dependencies.
# On a local machine this may take about 30 minutes, but only needs to be done once
cd /contur/docker/contur-visualiser
docker build -t contur-website .

# Your scan directory is assumed to have been download locally to $DATASET_DIR
# to enter the container, use the below, where −p 80:80 is exposing the correct port to the
# container, such that the visualisation page can be accessed from a local browser
docker run -it -p 80:80 -v {$DATASET_DIR}:/dataset contur-website

# Now, from within the container, run the contur−visualiser tool, providing, the x/y variables
# as well as the path to the dataset and .map file.
# if you do not have a .map file yet, you may use the usual contur −g $DATASET_DIR command first
cd contur-visualiser
./contur-visualiser -d {$DATASET_DIR} -x {X} -y {Y} -m <contur.map path>

# Once the visualiser has collected the contur−extract−herwig−xs−br output, open http://0.0.0.0:80/
# on your local browser to view the result.
```

excluded points, for example. To dig further into the details of a given point, the user can click on a point on the heatmap, and this will trigger the terminal in the DOCKER container to run `contur-mkhtml` on that point. Once the terminal has finished running that command, a further click will open a new window, displaying the summary plots for that point, similar to those shown in Figure 6. A detailed example of how to run the `contur-visualiser` tool is provided in Listing 23.

## C.11 Other tools

As explained in Section 4.1, the `contur-mkana` helps the user to generate static lists of available analyses to feed into the MCEG codes. A HERWIG-style template file is created in a series of `.ana` files, and a shell script to set environment variables containing lists of analyses useful for command-line-steered MCEGs is also written.

`contur-mkthy` is designed to help prepare SM theory predictions for particular RIVET analyses, for a more robust statistical treatment. This information is not always provided by the HEPDATA entry of a given measurement, so it sometimes has to be obtained from an alternative source. This script helps the user translate the raw prediction into a format usable by CONTUR. This tool is not usually intended to be needed by regular users.

## D BSM models as UFO files

The Universal FeynRules Object [13] (UFO) format is a Python-based way to encapsulate the Lagrangian of a new model. It contains the basic information about new couplings, particles and parameters which are required to generate BSM events. Since its inception in 2011, this format has become something of an industry standard, which is well-known and commonly used by theorists, and there exists a database of models with such implementations[6]. Furthermore, as the name implies, it is a format which is compatible with multiple event generators.

To get started with the study of a particular BSM model with CONTUR, the UFO file should

---

[6]https://feynrules.irmp.ucl.ac.be/wiki/ModelDatabaseMainPage

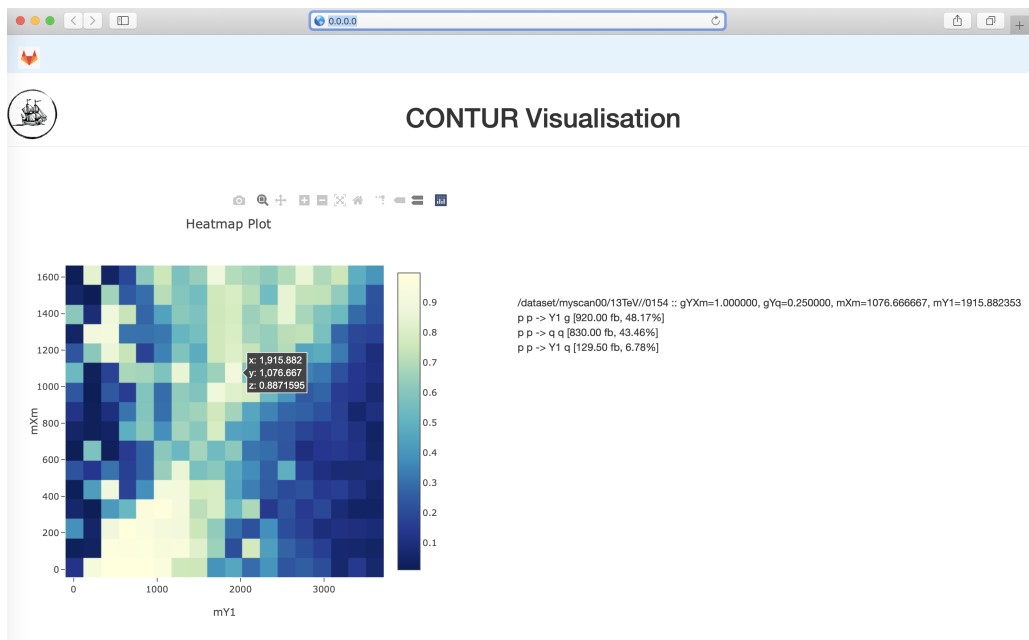

Figure 8: Screenshot showing the output of the `contur-visualiser` utility.

Listing 24: Using a directory of SLHA files as the basis for a scan.

```
[Parameters]
[[slha_file]]
mode = DIR
name = "/home/username/slha1_files"
```

be copied into the local `RunInfo` directory. The documentation associated with the model should give the adjustable parameter names. The precise next steps will depend upon the generator. A detailed example using HERWIG is provided in section Appendix B.

# E  Support for SLHA files

SUSY Les Houches Accord files provide a mechanism for supplying the mass and decay spectra of the particle content of a new given model, and for a number of widely used BSM scenarios which are already implemented in general purpose MCEGs, provide a convenient way of specifying the parameter point under consideration. CONTUR can make use of SLHA files in two ways. To do so it relies on the PYSHLA package [136].

## E.1  Scanning over a directory

If a set of parameter points is predetermined, and the SLHA files are available, these can be used simply as the input of a CONTUR scan. The `param_file.dat` syntax is as given in Listing 24.

To make the parameters from the SLHA file available for plotting in the produced `.map` file, use the `-S` parameter to pass a comma-separated list of SLHA block names to `contur` when running on the grid. The parameter names will have the block name prepended, for example `MASS:1000022` would be the mass of the lightest neutralino.

Listing 25: Using a single SLHA file as the basis for a scan.

```
[Parameters]
[[slha_file]]
mode = SINGLE
name = C1C1WW_300_50_SLHA.slha1
[[M1000022]]
block = MASS
mode = LIN
start = 10
stop = 210
number = 5
[[M1000024]]
block = MASS
mode = LIN
start = 200
stop = 300
number = 5
```

Listing 26: Scaling values in a single SLHA file as the basis for a scan.

```
[Parameters]
[[slha_file]]
mode = SCALED
name = RPV-UDD.slha
[[RVLAMUDD]]
mode = LIN
start = 1.00E-02
stop = 1.00E-01
number = 15
```

### E.2 Modifying SLHA files

If a single SLHA file is available and the user wishes to modify it, for example scanning two of the particle masses over some range, this can be done using the `param_file.dat` syntax given in Listing 25.

This would use the SLHA file `C1C1WW_300_50_SLHA.slha1` as a template, and would scan the $\chi_0$ (PID 1000022) and $\chi_1^\pm$ (PID 1000024) particle masses over the ranges specified by the mode, start, stop parameters in the specific number of steps. The modified parameters (only) would be written to the `params.dat` files for later use in analysis. A single letter in front of the particle ID integer is required, and should be unique to a given SLHA block, allowing (for example) several properties of the same particle (in different blocks) to be varied at once.

Alternatively, all the parameters in a single block may be scaled by a common factor, as shown in Listing 26, where the couplings in the RVLAMUDD block will also be multiplied by factors from 0.01 to 0.1, in 15 steps.

## F  Support for PANDAS DataFrames

PANDAS DataFrames provide an interoperable mechanism for supplying values for model parameters which are scoped into the CONTUR parameter sampler; and for a number of models bounded by other observations, imposing arbitrary constraints on parameter values.

For improved scan efficiency, `DataFrames` enable the generation of non-rectilinear grids and dynamic modification of scan resolution. CONTUR loads `DataFrames` from pickle files, which are files containing serialised Python object structures, and makes use of the PANDAS and PICKLE packages.

Listing 27: An example CONTUR configuration file showing the usage of a `DataFrame` to load a parameter

```
[Run]
generator = "/path/to/generatorSetup.sh"
contur = "/path/to/setupContur.sh"

[Parameters]
[[x0]]
mode = LOG
start = 1.0
stop = 10000.0
number = 15
[[x1]]
mode = CONST
value = 2.0
[[x2]]
imode = DATAFRAME
name = "/path/to/pickle_dataframe.pkl"
[[x3]]
mode = DATAFRAME
name = "/path/to/pickle_dataframe.pkl"
```

### F.1 Creating PICKLE files

If a PANDAS `DataFrame` with column names corresponding to parameter names in `param_file.dat` is available, `data_frame.to_pickle('path/to/file.pkl')` can be used to produce and save a PICKLE file to load into CONTUR.

### F.2 Loading PICKLE files

If a pickle file is available, the `DATAFRAME` block of a parameter file can be used to specify an absolute path, or path relative to the current working directory, under the keyword `name`, from which to load a pickle file into CONTUR. CONTUR only supports one pickle file for each `param_file.dat`, although an arbitrary number of parameters can be extracted from that file. An example is shown in Listing 27.

### F.3 Interoperability

The `DATAFRAME` mode can be used alongside other modes; for modes such as `LOG/LIN` with more than one parameter value, the scan will occur across each entry in the PANDAS `DataFrame`.

## G  Support for other event generators

### G.1  MADGRAPH support

Events are generated with MADGRAPH5_AMC@NLO [18] through a steering script, an example for which is given in Listing 28. This is functionally comparable to the `LHC.in` file for steering

HERWIG as shown in Listing 10. In the steering script, at first MADGRAPH-specific variables are being set. If a grid of signal points is to be generated using a batch system, it is important to include the options `set run_mode 0` and `set nb_core 1` as by default MADGRAPH runs on multiple cores which can be problematic on some HPC systems. These two lines configure MADGRAPH correctly for single core mode and make thus more efficient use of computational resources.

Listing 28: An example MADGRAPH steering script.

```
set group_subprocesses Auto
set ignore_six_quark_processes False
set gauge unitary
set loop_optimized_output True
set complex_mass_scheme False
set automatic_html_opening False
set run_mode 0
set nb_core 1
import model ./DM_vector_mediator_UFO
define p = g u c d s u~ c~ d~ s~ b b~
generate p p > t t~ Y1 DMS=2 QCD=4
output mgevents
launch
shower=Pythia8
set MY1 120
set MXm 130
set gYq 0.1
set gYXm 0.3
```

The desired UFO can be used by calling `import` ⟨*UFO model directory*⟩, which – in contrast to usage of HERWIG– does not need to be compiled. Afterwards, the model-specific processes are defined[7] and MADGRAPH started (`launch`). Parton showering for the generated events as well as giving a HEPMC file as output is taken care of by PYTHIA [137], initialised by `shower=Pythia8`. Afterwards, generation- and model-specific parameters are set. Just as for HERWIG, parameters should be included in curly brackets if CONTUR is used to generate a signal grid, otherwise concrete parameter values should be given.

After setting up the steering script, MADGRAPH generates events when called as `$MG_DIR/bin/mg5_aMC` ⟨*MG steering script*⟩ where `$MG_DIR` points to the installation directory of MADGRAPH. This will give HEPMC files as output in `mgevents/Events` that can be processed subsequently with RIVET to obtain a YODA file. Starting from this, the steps involving CONTUR are almost identical to those detailed for HERWIG in Sections B.4 to B.7. Due to different MC weight nomenclature within MADGRAPH, when running on a single parameter point, the option `--skip-weights` should be given to the `rivet` command as well as `--wn "Weight_MERGING=0.000"` to the `contur` command to ensure the correct MC weight is picked up by RIVET and CONTUR. A complete example for the steps from event generation with MADGRAPH to a CONTUR exclusion is given in Listing 29.

Listing 29: Steps from event generation with MADGRAPH to exclusion from CONTUR

```
$MG_DIR/bin/mg5_aMC mg_dmv.sh # generate events with MadGraph
rivet --skip-weights -a $CONTUR_RA13TeV \
    mgevents/Events/run_01/tag_1_pythia8_events.hepmc # run Rivet on generated events
contur Rivet.yoda --wn "Weight_MERGING=0.000" # get exclusion with Contur
```

To generate a signal grid with CONTUR using `contur -g`, specify MADGRAPH to be used as the MC generator by giving the option `--mceg madgraph`.

---

[7]In the example, the arbitrary choice of a top quark pair produced in association with the mediator is made.

## G.2 POWHEG support

Events can be generated in POWHEG in the `.lhe` format using the `pwhg_main` executable together with an input file called `powheg.input`. These events can then be transformed to the `.hepmc` format and showered using a full-final-state generator such as PYTHIA. These `.hepmc` events can then be passed through RIVET as usual to obtain a YODA file for processing by CONTUR to get exclusion limits.

Machinery to steer POWHEG using CONTUR has been created based on the PBZpWp POWHEG package which produces events at leading and next-to-leading order for electroweak $t\bar{t}$ hadroproduction in models with flavour non-diagonal $Z'$ boson couplings and $W'$ bosons [138, 139]. Three BSM models are currently implemented, namely the Sequential Standard Model (SSM) [140], the Topcolour (TC) model [141, 142], and the Third Family Hypercharge Model (TFHMeg) [143]. In what follows we exemplify this steering chain by explaining how to run jobs on a HPC system to set exclusion limits on the mass of $Z'$ in the SSM. POWHEG running does not support the UFO format, but the example discussed in this section could be used as an example if one wanted to use other POWHEG packages.

To run a batch job one needs three executables (`main-pythia`, `pwhg_main`, and `pbzp_input_contur.py`), two files (`param_file.dat` and `powheg.input_template`), and one directory (`RunInfo`), all in one run directory. `main-pythia` is responsible for the creation of the HEPMC file and of the parton showering. More details on these are listed below.

- The `RunInfo` directory contains the needed analysis steering files (`.ana`) and can be prepared as described in Section 4.4.

- The `pbzp_input_contur.py` script is used to create and fill the `powheg.input` files based on the model choice in `param_file.dat`, it needs `powheg.input_template` in order to do so.

- The `param_file.dat` file defines a parameter space, as with other generators.

In the SSM, there are only two parameters, *i.e.* the mass (`mZp`) and the width (`GZp`) of the $Z'$ boson in GeV, but one also needs to include the name of the model (SSM in this example), and the parameters of the other models as dummy[8]. The `param_file.dat` of the SSM should be formatted as in Listing 30, where `setupPBZpWp.sh` is a script which sets the environment needed to run `pwhg_main` and `setupEnv.sh` a script which sets up the run-time environment which the batch jobs will use, as a minimum it will need to contain the lines to execute your `rivetenv.sh` and `yodaenv.sh` files. For all the setup files, one should give the full explicit path. The `setupPBZpWp.sh` and the `setupEnv.sh` should be always in the same order as shown in this example, *i.e.* in `generator` one first gives the full path to `setupPBZpWp.sh` then the one for `setupEnv.sh`. In addition, one should check that the parameters defined in `params_file.dat` are also defined in `powheg.input_template`, in other words, removing or adding new parameters should be done in both files.

The HPC submission procedure using `contur-batch` follows the same workflow as for other MCEG options, but specifying `--mceg pbzpwp` and `-t powheg.input_template` to indicate the correct template. When the batch job is complete there should, in every run point directory, be a `runpoint_xxxx.yoda` file and an `output.pbzpwp` directory that contains the `.lhe` file. Creating the heatmap can then be done as explained in Sections B.6 and B.7.

---

[8]The angle $\theta_{sb}$ (`tsb`) needed for the TFHMeg, and $\cot\theta_H$ (`cotH`) needed for the TC model, since for now we only include the SSM, the TFHMeg and the TC models. This is done in order to be able to use the same `powheg.input_template` for all the models.

Listing 30: An example CONTUR configuration file for the SSM

```
[Run]
generator = "/path/to/setupPBZpWp.sh","/path/to/setupEnv.sh"
contur = "/path/to/setupContur.sh"

[Parameters]
[[mZp]]
mode = LIN
start = 1000.0
stop = 5000.0
number = 9
[[GZp]]
mode = LIN
start = 50.0
stop = 500.0
number = 10
[[model]]
mode = SINGLE
name = SSM
[[tsb]]
mode = SINGLE
name = dummy
[[cotH]]
mode = SINGLE
name = dummy
```

# H  The analysis database

The categorisation of RIVET analyses into pools, as described in Section 3.1 is implemented in an SQLITE database, distributed with CONTUR. The source code `analysis.sql` is in the `data/DB` directory, and after installation the compiled database will be in the same directory, named `analysis.db`. The database contains the following tables:

## H.1  General configuration

**beams**   Short text strings specifying known beam conditions, *e.g.* 13TeV.

**analysis_pool**   Defines the analysis pool names, associates them with a `beam`, and gives a short text description of the pool.

**analysis**   Lists the known RIVET analyses, assigns them to an `analysis_pool`, and stores the luminosity used, in the units corresponding to those used in the RIVET code.

**blacklist**   Optionally, for a given `analysis`, defines any histograms (via regular expression matching) which should be ignored.

**whitelist**   Optionally, for a given `analysis`, defines any histograms (via regular expression matching) which should be used. If an `analysis` has any whitelist entries, all unmatched histograms will be ignored.

**subpool**    Optionally for a given `analysis`, list (and name) subsets of histograms which are know to be statistically "orthogonal" in the sense of containing no events in common.

**normalization**    Some measurements are presented as area-normalised histograms (for example when the discussion focuses on shapes). CONTUR requires the cross section normalisation, so that it knows the weight with which signal events should be injected. For such histograms, this table stores this normalisation factor. For searches, where the measured distribution is often just a number of events per (number of units), this results sometimes in bins with unequal width. In this case, the "number of units" should be given in the `nxdiff` field. The number of events in each bin will be obtained by multiplying by the bin width and dividing by `nx`. If the bin width is constant, this can be left as zero, and will not be used.

**needtheory**    Analyses which both require and use the SM prediction.

## H.2   Special cases

The remaining tables define various special cases of analyses which may be included or not in a CONTUR run by setting command-line options at run-time. See Section 3.3 for usage and more discussion on why these special cases are treated differently.

**metratio**    Missing energy ratio measurement(s) from ATLAS. Included by default.

**higgsgg**    $H \to \gamma\gamma$ analyses. Included by default.

**searches**    Search analyses (for which detector smearing is used). Excluded by default.

**higgsww**    $H \to WW$ analyses. Excluded by default.

**atlaswz**    ATLAS $WZ$ analysis. Excluded by default.

**bveto**    Analyses with a $b$-jet veto which is not implemented in the fiducial phase space. Excluded by default.

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
