# Peer review of "Testing new-physics models with global comparisons to collider measurements: the Contur toolkit"

_SciPost Physics Core, doi:SciPost Phys. Core 4, 013 (2021)_

## Round 1 · Referee Report · Frank Siegert · 2021-3-5

Strengths
1) The publication describes a publically available software tool and thus has merits beyond the immediate content of the document itself.
2) Very accessible and pleasant to read introduction to the guiding principles, strengths, and limitations of the Contur approach.
3) The review/library of tricky aspects of LHC analyses in Sec. 3.3 is very valuable, even if it is just a side note of the manuscript.
Weaknesses
There are no significant weaknesses in the manuscript, only very few and minor clarifications needed (and requested as changes below).
But given that I consider the availability of Contur as public tool a major strength of the manuscript, I would like to remark one small weakness in that tool itself:
1) Getting started with the Contur program as a new user, one hits a few blocks on the road. Either because documentation is not complete, outdated, or small bugs make the simple example run in the tutorial fail at various stages. These are no major obstacles though and I think the authors can improve these aspects very quickly. With some workarounds to these problems I have been able to get my first Contur exclusion results within ~1 h.
Report
This manuscript describes an original new approach to searches for new physics through existing collider measurements. It embraces the current situation of particle physics, with no clear theory guidance towards BSM scenarios and thus the need for a broad and model-independent data-driven approach.
The algorithm or program is not completely novel, but its new availability as a public tool makes the manuscript valuable as a description of the physics and manual of the technicalities.
I am happy to see this published and only have a few small change requests for clarifications in the text regarding points which were not completely clear to me during the initial reading.
One question out of interest (no requirement regarding the document): Can the scan utility be extended to iteratively find the least constrained parameter points? Maybe using similar techniques to tuning tools like Professor/Apprentice?
Requested changes
Please let me know if I misunderstood any of the following or else clarify the following points in the manuscript:
1) You describe the operating mode where only the BSM MC is produced, and the SM background is either taken from HepData or the data used as a proxy. It is clear how this "stacking" works for differentical cross sections. But how is this achieved for other measurements, for example even normalised cross sections can't be used because you can't simply add the BSM MC a posteriori, right? Let alone profile histograms or similar objects?
2) In Sec. 3.1: I was a bit confused when reading the part about "... final states arising from essentially the same events ..." is this referring to correlations between *data* events within different measurements, or *BSM MC* events populating different observables/regions simultaneously? I assume the former, but might be nice to phrase explicitly.
3) You only include LHC analyses. Would non-LHC analyses provide no improvement, or is that a pragmatic decision because their documentation/Rivet analyses are often not rigorous enough?
4) Since the document has the form of a manual, it would be good to refer the user to the README.md in git for setup instructions to quickly get started. Currently this is not mentioned in the publication nor referenced on the homepage, but is the only place that describes how to install Contur.
Even better would be to unify all documentation. Currently there are at least three documentation starting points for the user: This manuscript, the README.md, and the homepage's "Using Contur".
5) Minor glitches in the Contur tool/documentation to be resolved: https://gitlab.com/hepcedar/contur/-/issues?state=all&author_username=fsiegert

---

## Round 1 · Referee Report · Anonymous · 2021-3-5

Report
Reinterpretation of the highly-valuable LHC data and search is a pressing subject given the rapidly growing number of dedicated searches and their sophistication. The theory development in particle physics requires understanding existing constraints to infer the valid parameter space and motivating new searches. This manuscript describes the newly developed contour v2 package to reinterpret the LHC results using the LHC validated Rivet analysis preservation Library.
The manuscript provides a clear description of the workflow, the Rivet Library, as well as sampling, evaluation of likelihood, and visualization of the model parameter space. It also provides sample code and useful examples in the appendix.
The research is on the critical topic of understanding the LHC results, with concrete results and developments, and is well-written. I want to recommend it for publication.

---

## Round 1 · Referee Report · Sezen Sekmen · 2021-3-14

Strengths
1. This paper describes in detail Contur, a tool that makes it possible to determine a proposed new physics model's consistency with the standard model by comparing the new model's predictions with experimental measurements recorded in the well-established tool Rivet. The Contur method is unique and original, and fills a much needed gap in the area of interpreting experimental results in terms of theoretical predictions.
2. The paper describes Contur v2, a version that is made available for use by everyone, not only by a limited number of experts. The authors' effort to make the tool publicly usable is appreciated.
3. The paper goes into a lot of detail to systematically introduce all functionalities of Contur.
4. A concrete physics example is shown in the appendix to demonstrate how Contur works and what results can be obtained from it.
5. Statistical method of inference about a new physics model is described clearly in detail.
Weaknesses
1. The paper gives a lot of details, however the way it is structured makes it difficult for a potential user to get a basic overview of how Contur is structured.
2. The boundary between Rivet and Contur is not always clear.
3. It is overall not clear which Contur functionalities are controlled by the user and which are the functionalities belonging to the internal workflow of the code.
4. Instructions for a standard installation and a basic test run are missing. Instructions exist for a docker-based run for full case, but they are distributed in several sections in the main text and appendices.
Report
I would recommend this manuscript for publication here after the text has been modified to present a more practical description to the users by addressing the comments below.
Requested changes
1. In the main body, provide a technical workflow of Contur following the conceptual workflow in Section 2.1. A flowchart depicting the technical workflow will be very useful. The flowchart could include input files, output files and packages/routines that process and produce those.
2. Related to the previous point: Provide the source code repository already in the main body of the text. The main body starts with conceptual physics descriptions, but especially starting with section 4, a lot of names for Contur and Rivet functionalities are referenced. It is difficult to follow these for a first time reader without understanding their position in the Contur or Rivet packages.
3. Please also better clarify the task division between Contur and Rivet, i.e. in the flowchart. What is done by either package is mentioned in various places in the text, but it would help to have a concrete, dedicated description.
4. In the main body, it is not always clear which Contur functionalities are controlled by the user and which are the functionalities belong to the internal workflow of the code.
5. It would be helpful to present the content of an example YODA file. I understand that the YODA files are generated both for the experimental data by Rivet and for the BSM models by Contur via Rivet. Is that correct? Please clarify.
6. Figure 3: Please improve the caption and the legends. What are the red histograms? What parameters do the numbers in right and left histograms corresponding to the red histograms describe?
7. Sec 6.2.2. What is the purpose of adding theory functions? Providing an example physics case could be useful.

---

## Round 2 · Referee Report · Frank Siegert · 2021-4-30

Report

I am happy with the new version and how the authors have addressed my previous comments. Thanks also for fixing the problems I encountered when running the tutorial so quickly!

---

## Round 2 · Referee Report · Anonymous · 2021-5-3

Report

I am happy with the changes made by the authors based on my previous comments. I recommend the manuscript for publication.

---

## Round 2 · Author Response

We thank all the referees and the editors for their comments on the manuscript. We have submitted a revised versions which addresses the points raised, and which we believe have collectively improved the manual.

----------- Referee #1 -----------

We thank Referee #1 for reviewing our paper and for their report. They did not raise any specific concerns.

----------- Referee #2-----------

We are grateful to Referee #2 for their comments, for trying out the tutorial included in the paper, and for finding remaining bugs and issues, which were logged as issues on our gitlab repository. We have fixed these issues and therefore hope that future readers will have no problem following the tutorial.
Referee #2 asked a number of specific question, which we will respond to here:

>>One question out of interest (no requirement regarding the document): Can the scan utility be extended to iteratively find the least constrained parameter points? Maybe using similar techniques to tuning tools like Professor/Apprentice?

Response: Regarding extending the scanning tool to locate the least constrained parameter point: this can be done by making use of the CONTUR code interface and connecting to a numerical scanner or minimiser. There are myriad extensions (and issues) in doing so, for example the existence of many minima rather than a unique point. More efficient scanning of multi-dimensional parameter spaces is an area of active research in the CONTUR team and in the reinterpretation community more widely. We now have added a short paragraph about these possible extensions in the manual, at the end of section 4.2

>>You describe the operating mode where only the BSM MC is produced, and the SM background is either taken from HepData or the data used as a proxy. It is clear how this "stacking" works for differentical cross sections. But how is this achieved for other measurements, for example even normalised cross sections can't be used because you can't simply add the BSM MC a posteriori, right? Let alone profile histograms or similar objects?

Response: Normalised histograms are complemented with a fiducial cross-section factor in the analysis database (when this is provided by the experiment), which allows rescaling to the differential cross-section for addition, then re-normalising. Ratio plots have a similar special treatment, and a profile histogram treatment is being developed in the same way. We have added some comments about this at the end of Sec 3.0 in the manual.

>> In Sec. 3.1: I was a bit confused when reading the part about "... final states arising from essentially the same events ..." is this referring to correlations between *data* events within different measurements, or *BSM MC* events populating different observables/regions simultaneously? I assume the former, but might be nice to phrase explicitly.

Response: You are right, this refers to measurements where the data which enter the selection may be partially the same. We have rephrased the statement to resolve the ambiguity.

>> You only include LHC analyses. Would non-LHC analyses provide no improvement, or is that a pragmatic decision because their documentation/Rivet analyses are often not rigorous enough?

Response: In principle this method could be extended to non-LHC analyses, although we would need to add additional running modes for the collision energies etc. We have added a statement about this at the end of Sec 3.1 in the manual

>> Since the document has the form of a manual, it would be good to refer the user to the README.md in git for setup instructions to quickly get started. Currently this is not mentioned in the publication nor referenced on the homepage, but is the only place that describes how to install Contur. Even better would be to unify all documentation. Currently there are at least three documentation starting points for the user: This manuscript, the README.md, and the homepage's "Using Contur".

Response: The reason that we did not directly link to the gitlab page in the manual is that there is always a risk that at some point in the future, the code may migrate from gitlab to elsewhere. Therefore, this may not be a 100% stable link forever. Instead, we propose to link to the CONTUR homepage in the introduction and summary,, which then links to the most-recent documentation, setup instructions, and codebase.

>> Minor glitches in the Contur tool/documentation to be resolved: https://gitlab.com/hepcedar/contur/-/issues?state=all&author_username=fsiegert

Response: The glitches pointed out by the referee have all been resolved.

----------- Referee #3-----------

We thank the referee for their careful reading of the paper and helpful comments, which we have addressed to make the manual easier to digest. In particular, we feel like comments 1-3 in the "Weaknesses Section" can be achieved with a detailed overview diagram, which we have created and added to the appendix. This diagram gives an overall view of the CONTUR workflow and ecosystem, while pointing out who (user, external tool or CONTUR [and if so, which executable]) is responsible to execute each step. We hope this clarifies these points. For item 4, the example in the appendix “Example Contur study with Herwig“ is an end-to-end example of the single-point workflow. We do not propose to put detailed installation instructions in the manual - these may become out of date over time! Instead, we propose to link the CONTUR homepage which points the user to the latest source code and setup instructions. We hope the reviewer is satisfied with this suggestion.

We reply to specific points raised by the Referee #3 below.

>> 1. In the main body, provide a technical workflow of Contur following the conceptual workflow in Section 2.1. A flowchart depicting the technical workflow will be very useful. The flowchart could include input files, output files and packages/routines that process and produce those.

Response: We thank the referee for their suggestion and we agree this would be very helpful. Such a diagram is now provided in App A, showing all steps of the workflow, in/out files and their formats, and which executables or (external) tools to use at each step.

>> 2. Related to the previous point: Provide the source code repository already in the main body of the text. The main body starts with conceptual physics descriptions, but especially starting with section 4, a lot of names for Contur and Rivet functionalities are referenced. It is difficult to follow these for a first time reader without understanding their position in the Contur or Rivet packages.

Response: The reason that we did not directly link to the gitlab page in the manual is that there is always a risk that at some point in the future, the code may migrate from gitlab to elsewhere. Therefore, this may not be a 100% stable link forever. Instead, we propose to link to the CONTUR homepage in the introduction and summary, which then links to the most-recent documentation, setup instructions, and codebase. In combination with the flowchart described above, we hope this gives a clear guide to each step and the technical terms.

>> 3. Please also better clarify the task division between Contur and Rivet, i.e. in the flowchart. What is done by either package is mentioned in various places in the text, but it would help to have a concrete, dedicated description.

Response: We now address this in the flowchart.

>> 4. In the main body, it is not always clear which Contur functionalities are controlled by the user and which are the functionalities belong to the internal workflow of the code.

Response: We now address this in the proposed flowchart by specifying which executable to use at each step, or whether an external tool is used.

>> 5. It would be helpful to present the content of an example YODA file. I understand that the YODA files are generated both for the experimental data by Rivet and for the BSM models by Contur via Rivet. Is that correct? Please clarify.

Response: Since YODA files are a standard output of Rivet and are one of the possible output formats of HEPData, and are not specific to CONTUR, the authors do not think this is the right place to exhaustively document their contents. Nonetheless, we added in Section 3 a few brief lines which give more information about the current YODA format, for the benefit of the reader.

---

## Round 2 · List of Changes

- At the end of Section 1 (Introduction) and also in the Summary, we added a pointer to the CONTUR homepage, which will host the latest installation instruction and links to the source code.
- In the second paragraph of Section 2 (Overview), we now provide a sentence highlighting the existence of the detailed flowchart in Appendix A. This is also pointed out at the end of the second paragraph of Sec 2.1, and in the caption of Fig1.
- In Section 3 (Rivet analyses), at the end of the first paragraph, we give a few lines of explanation about the YODA format. Also in this section, we updated the list of Rivet reference analyses since more have come out since the original submission of the manuscript, and they are now included in CONTUR.
- At the end of Sec 3.0, we added a short discussion on how BSM contributions are stacked for other data types than differential cross-sections.
-At the end of Sec 3.1, we remake on how one could in principle use the CONTUR framework for results from other experimental facilities such as LEP or HERA.
- At the end of Sec 4.2, we remark upon how the CONTUR scanning machinery could be extended.
- Fig 3 caption has been rewritten in accordance with the requests of the Referees.
- Fig4 has been updated to account for the latest measurements which have been made available in Rivet/CONTUR.
- Sec 6.2.1/6.2.2: As pointed out by the referees, the "Data functions" and "Theory functions" functionality was confusingly named. We have renamed them as " Plotting external grids" and "Plotting external functions" respectively, in the code and in the manual. Each has a small amount of additional content in those sections to clarify their use.
- Added new Appendix A/Fig 5: flowchart

---

## Editorial Decision

published